# Continual Learning on a Data Diet

## Abstract

Continual Learning (CL) methods usually learn from all the available data. However, this is not the case in human cognition which efficiently focuses on key experiences while disregarding the redundant information. Similarly, not all data points in a dataset have equal potential; some can be more informative than others. Especially in CL, such redundant or low-quality data can be detrimental for learning efficiency and exacerbate catastrophic forgetting. Drawing inspiration from this, we explore the potential of learning from important samples and present an empirical study for evaluating coreset selection techniques in the context of CL to stimulate research in this unexplored area. We train different continual learners on increasing amounts of selected samples and elucidate the learning-forgetting dynamics by shedding light on the underlying mechanisms driving their improved stability-plasticity balance. We present several significant observations: learning from selectively chosen samples (i) enhances incremental accuracy, (ii) improves knowledge retention of previous tasks, and (iii) continually refines learned representations. This analysis contributes to a deeper understanding of selective learning strategies in CL scenarios. The code is available at `https://anonymous.4open.science/r/Data-Diet-CD87`.

## 1 Introduction

Machine learning has achieved remarkable success in solving complex tasks, often relying on the assumption that data is available in a static and complete form. In traditional machine learning, models are trained once on fixed datasets and are evaluated without further updates. While effective in controlled scenarios, this approach falls short in dynamic environments where data and tasks evolve over time. Addressing this limitation requires a shift from static learning paradigms to more adaptive systems capable of learning continuously. Continual Learning (CL) bridges this gap by enabling models to learn from a stream of data sequentially. Unlike traditional machine learning, CL emphasizes retaining previously acquired knowledge while learning new tasks, mimicking the way humans accumulate and adapt knowledge over their lifetimes. Class-Incremental Learning (CIL) is the most challenging scenario of CL where the learner is required to predict outcomes for all encountered classes without being given task identifiers (Van de Ven & Tolias, 2019). However, catastrophic forgetting (McCloskey & Cohen, 1989) remains a challenge in this dynamic setting wherein the class-incremental learners tend to lose acquired knowledge from previous tasks, upon learning new ones. Recent research has brought solutions through various techniques, including regularization methods (Kirkpatrick et al., 2017; Li & Hoiem, 2017; Lee et al., 2017), replay strategies (Chaudhry et al., 2018; Lopez-Paz & Ranzato, 2017; Aljundi et al., 2019; Borsos et al., 2020), architecture expansion (Yan et al., 2021; Wang et al., 2022a; Zhou et al., 2022; Rusu et al., 2016; Yoon et al., 2019) and prompt learning (Wang et al., 2022c;b; Smith et al., 2023) approaches. However, these approaches aim to learn from all the available data during training to maximize model performance and assume that all samples are equally important. This standardized practice may not fully reflect the efficiency and adaptability observed in human learning since, as humans, we are initially exposed to vast amounts of information but intuitively filter and prioritize them, focusing on key experiences (e.g. clear and novel examples) that enrich our understanding while disregarding redundant details (Pagnotta et al., 2022; Jones et al., 2016; Posner & Petersen, 1990).

We draw inspiration from this human cognitive ability and introduce an empirical study to evaluate the learning-forgetting dynamics of different CIL models when trained with important samples selected by a wide range of sample selection approaches (as illustrated in Figure 1). Through a detailed analysis, we provide insight into how data selection leads to an improved stability-plasticity balance in continual learning.

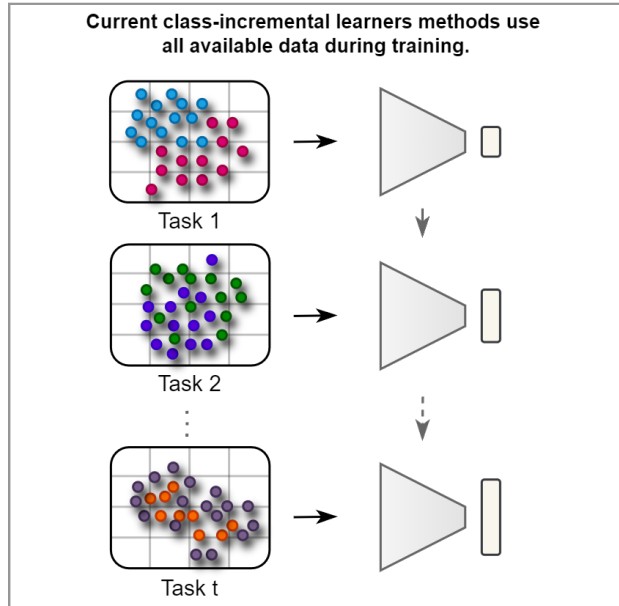 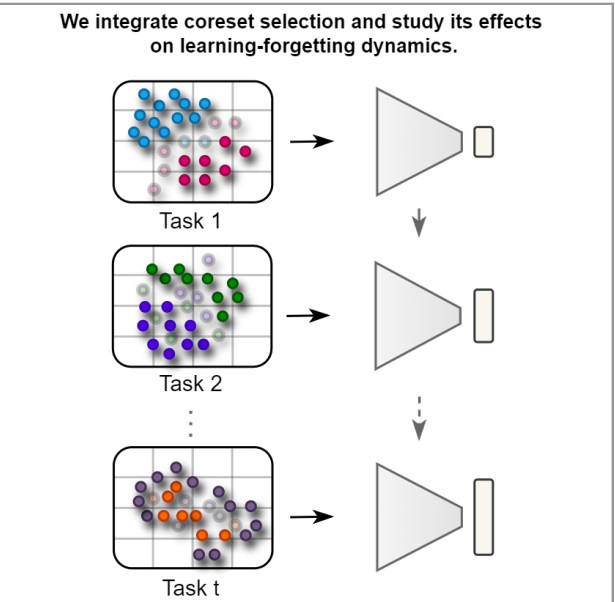

Figure 1: Illustration of our evaluation protocol: Existing class-incremental learning methods (**left**) typically utilize all available samples indiscriminately during training. In this study (**right**), we subject class-incremental learners to a *data diet* and analyze how the selection of the most important samples with different coreset selection methods affects the incremental performance.

We believe that this comprehensive study and investigation contributes to a deeper understanding of the potential benefits of sample selective learning strategies in CIL scenarios and stimulates systematic research that leverages these insights to take a more holistic and data-centric approach to continual learning.

Our contributions can be summarized as:

I. This paper presents the first explicit empirical analysis of different coreset selection methods in combination with various continual learners in the class-incremental learning setting.

II. We find that learning from selectively chosen samples with different coreset selection methods significantly elevates incremental learning performance.

III. We demonstrate that the increase in performance among class-incremental learners trained with selected samples arises from enhanced retention of previously acquired concepts due to improved representation and perception of the models.

IV. We show that continual learning can benefit from a data-centric approach, despite the fact that most existing research has predominantly focused on model-centric enhancements.

## 2 Background

### 2.1 Class-Incremental Learning

Class-incremental learning can be broadly categorized into four main approaches (Van de Ven & Tolias, 2019); regularization, replay, architecture-based and prompt-based. Regularization-based methods regularize the abrupt changes in the learned parameters to prevent catastrophic forgetting (Kirkpatrick et al., 2017; Li & Hoiem, 2017; Lee et al., 2017). Replay-based methods either retain selected exemplars from prior tasks or generate a subset of data points from previous tasks to alleviate forgetting (Chaudhry et al., 2018; Lopez-Paz & Ranzato, 2017; Aljundi et al., 2019; Borsos et al., 2020). Architecture-based methods prevent forgetting by increasing model size and allocating distinct sets of parameters to individual tasks, ensuring there is no

overlap between them (Yan et al., 2021; Wang et al., 2022a; Zhou et al., 2022; Rusu et al., 2016; Yoon et al., 2019).Recently, with the growing popularity of large pretrained models with Vision Transformers (ViT), prompt-based methods also received growing popularity (Wang et al., 2022c;b; Smith et al., 2023).

**Summary of CIL Methods Selected for Analysis**

We use 7 well-established CIL models that encompass various approaches including architecture-based, replay-based, regularization-based, and prompt-based. We deliberately chose these methods to provide a comprehensive analysis since they all represent different learning strategies. For more details, please see our Appendix A.1.

**DER-Architecture.** Dynamically expandable representation (Yan et al., 2021) creates a new backbone (neural network) as a feature extractor for each task and then aggregates the features of all backbones on a single classifier. Each new or expanded backbone uses an additional auxiliary loss to differentiate better between old and new classes. Facing new tasks, it freezes the old backbone to maintain former knowledge.

**FOSTER-Architecture.** Feature boosting and compression for class-incremental learning (Wang et al., 2022a) frames the learning process as a feature-boosting problem and aims to enhance the learning of new features. Then, it expands the continual learner on a single classifier by integrating the boosted features with a compression step to ensure that only relevant features are retained.

**MEMO-Architecture.** Memory efficient expandable model (Zhou et al., 2022) expands the network in a more efficient way. It assumes that the initial blocks of the backbone capture the general patterns for any task and only expands the model in the last or specialized blocks that are designed to be task-specific.

**iCaRL-Replay.** Incremental Classifier and Representation Learning (Rebuffi et al., 2017) is a replay-based method that stores samples from each learned task. Upon the arrival of a new task, it uses stored exemplars together with the new one to capture the distribution at once. Therefore, it refines the features after each task with additional distillation loss to overcome abrupt shifts in the feature space.

**ER-Replay.** Experience Replay (Rolnick et al., 2019) is a simple yet strong method that employs reservoir sampling to store samples from each task and randomly retrieves stored samples with the new task to capture the distribution all at once.

**LwF-Regularization.** Learning without Forgetting (Li & Hoiem, 2017) is solely a regularization-based method without relying on any replay buffer. It utilizes a distillation loss to prevent sudden changes in the feature space while learning new tasks.

**CODA-Prompt.** CODA-Prompt (Smith et al., 2023) as the name suggested is a prompt-based method that leverages pretrained Vision Transformers (ViT) without relying on data rehearsal. It introduces a set of prompt components that are dynamically assembled based on input-conditioned weights, generating task-specific prompts for the transformer's attention layers. These generated prompts selectively guide the model's attention to relevant features for each task, to enable better stability-plasticity tradeoff.

## 2.2 Coreset Selection

Coreset selection approximates the distribution of the whole dataset with a small subset and has been extensively examined in data-efficient supervised batch learning (Toneva et al., 2018; Guo et al., 2022; Coleman et al., 2019a; Paul et al., 2021; Welling, 2009; Coleman et al., 2019b; Iyer et al., 2021; Mirzasoleiman et al., 2020) and active learning (Wei et al., 2015; Sener & Savarese, 2017). Coreset selection also holds promise in continual learning to construct a memory buffer from important samples (Aljundi et al., 2019; Borsos et al., 2020). Recently, an inspiring study (Yoon et al., 2022) improved the performance in online CL setup by introducing a coreset selection method to select the most diverse samples while approximating the mean of a given batch.

However, besides this one method (Yoon et al., 2022), the interplay between coreset selection methods and continual learning models remains unexplored. This warrants deeper investigation into their interaction as well as the underlying mechanisms related to the improved performance. Exploring this interaction, by focusing on the quality of the data itself, could provide novel insight to create more efficient and advanced continual learners.

**Overview of Coreset Algorithms Selected for Analysis**

We employ 4 distinct coreset selection methods as well as a baseline using random selection. Once again, we carefully chose these distinct methods to offer comprehensive empirical analysis. It is important to note that these coreset selection methods require a brief initial training or warm-up phase to make informed and meaningful decisions when selecting coreset samples. We provide more details in our Appendix A.2.

**Random.**  This selection strategy involves randomly selecting a subset of data points from the entire dataset without any specific criteria or consideration of their importance or informativeness.

**Herding.**  Herding (Welling, 2009) chooses data points by evaluating the distance between the center of the original dataset and the center of the coreset within the feature space. This algorithm progressively and greedily includes one sample at a time into the coreset, aiming to minimize the distance between centers.

**Uncertainty.**  Samples with lower confidence levels might have a stronger influence than those with higher confidence levels, thus having these samples in the coreset can be useful. Least confidence, entropy, and margin are the common metrics used to quantify sample uncertainty (Coleman et al., 2019b). In this study, entropy is used as a selection metric.

**Forgetting.**  Forgetting selects instances which were correctly classified in one epoch and then subsequently misclassified in the following epoch during training (Toneva et al., 2018). This method provides valuable insight into the intrinsic characteristics of the training data and removes challenging or forgettable instances.

**GraphCut.**  GraphCut partitions the dataset into subsets based on dissimilarity or information content, and data points from these subsets are then selected to form the coreset (Iyer et al., 2021). This approach ensures that the coreset captures the diversity and essential information of the original dataset while reducing redundancy.

## 3 Data Diet

We conduct a comprehensive evaluation of existing CIL methods, assessing their performance when trained on purposefully selected, informative samples, as opposed to the traditional approach of full dataset training. We refer to this as a 'Data Diet'. To clarify our approach, we first present the necessary preliminaries and problem formulation in Section 3.1. Following this, we define our objective and outline the proposed training strategy in Section 3.2.

### 3.1 Preliminaries and Problem Formulation

Formally, we define the CIL problem as a sequence of classification tasks $T_{1:t} = (T_1, T_2, ..., T_t)$. Each task $T_t$ is drawn from an unknown distribution and consists of input pairs $(x_{i,t}, y_{i,t}) \in X_t \times Y_t$ where $x_{i,t}$ represents the sample and $y_{i,t}$ indicates the corresponding label. Note that these learning tasks are mutually exclusive, meaning that the label sets do not overlap, i.e., $Y_{t-1} \cap Y_t = \emptyset$.

From the coreset selection perspective, the aim is to find the most informative subset $S_t$ from a given task $T_t$ with a large number of input pairs $(x_{i,t}, y_{i,t})$. Therefore, model trained with subset $S_t \subset T_t$ with a condition of $|S_t| < |T_t|$ should have a similar generalization performance compared to a model trained with $T_t$.

### 3.2 Objective and Training Strategy

We structure the training process into two distinct phases: the warm-up phase and the learning phase. This is necessary because coreset selection methods operate by analysing how models behave and represent new data. Hence, CL models needs to be at least partially trained during the initial warm-up phase to identify the most informative samples for a given task correctly. It is important to note that the duration of the warm-up phase is typically much shorter than that of the learning phase. Upon completion of the warm-up phase, the learning phase proceeds with the selected subset of samples.

---

**Algorithm 1** CL on Data Diet

---

**Require:** Model $f_\theta$, Tasks $T_{1:t}$ with training sets $T_t$, learning rate $\eta$, total epochs e, warm-up fraction $\alpha$, coreset selection function $\phi$, coreset fraction $s$

1: **for** task t = 1 to $T_t$ **do**
2:     **for** epoch = 1 to $\lfloor \alpha e \rfloor$ **do**                                                           ▷ Warm-up Phase
3:         **for** each batch $b$ in $T_t$ **do**
4:             Compute $\mathcal{L}_{CE}(f_\theta, b)$
5:             Update $f_\theta \leftarrow \theta - \eta \nabla_\theta \mathcal{L}_{CE}$
6:         **end for**
7:     **end for**
8:     Use $\phi(f_\theta, T_t)$ to select $S_t \subset T_t$ with a fraction of $s$
9:     **for** epoch = 1 to $\lfloor (1-\alpha)e \rfloor$ **do**                                        ▷ Learning Phase
10:        **for** each batch $b$ in $S_t$ **do**
11:           Compute $\mathcal{L}_{\mathrm{CL}}(f_\theta, b)$
12:           Update $f_\theta \leftarrow \theta - \eta \nabla_\theta \mathcal{L}_{CL}$
13:        **end for**
14:     **end for**
15: **end for**

---

Let $f_\theta(\cdot)$ denote the continual learning model with parameters $\theta$. Then, the training process can then be expressed as follows:

$$f_{\theta^*} = \arg\min_\theta \mathcal{L}_{CL}(f_\theta, S_t, (1-\alpha)e) \circ \arg\min_\theta \mathcal{L}_{CE}(f_\theta, T_t, \alpha e) \tag{1}$$

Here, the second term $(f_\theta, T_t, \alpha e)$ represents training the model $f_\theta$ on the full training samples of task $T_t$ with a defined time budget of $\alpha e$ where hyperparameter $\alpha \in (0, 1)$ and determines the fraction of the total training budget allocated to the warm-up phase, and e is the total number of epochs available for training. Similarly, the first term $(f_\theta, S_t, (1-\alpha)e)$ represents the training of the model $f_\theta$, for the remaining time budget $(1-\alpha)e$, on the coreset $S_t$ which is selected from $T_t$ with a fraction of $s \in (0, 1)$ based on a coreset selection function $\phi(\cdot)$, so that $|S_t| = s \cdot |T_t|$. Note that $\mathcal{L}_{CE}$ represents Cross-Entropy loss and $\mathcal{L}_{CL}$ represents the loss defined by continual learning methods given in section 2.1.

To provide a more precise explanation, Algorithm 1 begins with a warm-up phase (lines 2-7) where the model $f_\theta$ observes the training samples $T_t$ of the current task for a duration of $\alpha e$. During this phase, the model trains each batch $b$ to compute the Cross-Entropy loss $\mathcal{L}_{CE}(f_\theta, b)$. This initial exposure allows the model to capture a broad understanding of the task's characteristics.

Following the warm-up (line 8), the algorithm employs the coreset selection function $\phi(\cdot)$ which requires training samples for a given task $T_t$ and the model $f_\theta$ to filter down to a coreset $S_t \subset T_t$, consisting of only a fraction $s$ of the current task samples. The criterion for selection, depending on the coreset selection function, can target samples with high informativeness, uncertainty, or relevance, focusing on key data points.

In the learning phase (lines 9-14), which spans the remaining $(1-\alpha)e$ epochs, the model is trained on batches from $S_t$, using specific loss function of continual learners $\mathcal{L}_{\mathrm{CL}}(f_\theta, b)$. This refines the goal of solidifying task-specific knowledge while minimizing interference from previous tasks to prevent catastrophic forgetting.

## 4 Experimental Setting

**Datasets.** We use well-established continual learning datasets, specifically **Split-CIFAR10** and **Split-CIFAR100** (Krizhevsky et al., 2009), **Split-ImageNet-100** (Russakovsky et al., 2015) in our experiments to evaluate and posit our findings. **Split-CIFAR10** has 5 disjoint tasks and each task has 2 disjoint classes with 10000 samples for training and 2000 samples for testing. **Split-CIFAR100** has 10 disjoint tasks and each task has 10 disjoint classes with 5000 samples for training and 1000 samples for testing. In addition, we employ **Split-ImageNet100**, a subset of the large-scale ImageNet dataset, with images at a higher resolution of 224x224 pixels. Similar to Split-CIFAR100, Split-ImageNet100 is divided into 10 tasks, each consisting of 10 disjoint classes. The increased number of classes, fewer images per class combined with longer learning sessions, and higher resolution bring further challenges and offer a more complex scenario.

**Implementation Details.** We use Deepcore (Guo et al., 2022) for coreset selection methods and PYCIL (Zhou et al., 2023) for the CIL. We employ both from scratch (ResNet18) and pretrained (ResNet18 and ViT) backbones with prior knowledge to provide a more comprehensive analysis, using standard CL metrics which are discussed more in detail in the Appendix A.3. We set the total training budget $e$ to 100 epochs where warmup fraction $\alpha$ is set to 0.1 and the remaining is allocated for the learning phase. We set coreset fraction $s$ to $10\%, 20\%, 50\%, 80\%$ and $90\%$ for each task. We use SGD optimizer with a scheduled learning rate of 0.1 and momentum of 0.9. We set a weight decay of $5 \times 10^{-4}$ for the initial task and $2 \times 10^{-4}$ for subsequent tasks. We set the batch size to 128. We employ a fixed memory size: 50 per class for CIFAR10 and 20 per class for CIFAR100 and ImageNet100. We do not employ coreset selection methods to construct the memory buffers, adhering instead to the original implementations. For ViT, we only modify the learning rate to 0.001, reduce the batch size to 32, and train for 20 epochs. We run experiments on A100 GPU with different seeds and report their average accuracy and standard deviation across three runs.

Table 1: Accuracy [%] of CIL models across various coreset fractions and selections on **Split-CIFAR10**. Learning from coreset samples enhances the performance, except FOSTER and LwF. The best results are highlighted in bold if coreset selection outperforms training with all samples.

| | Fraction | 10% | 20% | 50% | 80% | 90% | 100% |
|---|---|---|---|---|---|---|---|
| | Random | $51.79 \pm 4.6$ | $54.28 \pm 3.8$ | $55.68 \pm 0.3$ | $57.27 \pm 2.9$ | $55.61 \pm 2.5$ | $56.91 \pm 1.3$ |
| | Herding | $41.65 \pm 2.2$ | $52.35 \pm 2.5$ | $59.79 \pm 1.8$ | $63.96 \pm 1.1$ | $62.93 \pm 1.2$ | $56.91 \pm 1.3$ |
| DER (Yan et al., 2021) | Uncertainty | $56.02 \pm 1.7$ | $59.48 \pm 1.7$ | $57.97 \pm 0.8$ | $62.01 \pm 3.1$ | $59.36 \pm 1.5$ | $56.91 \pm 1.3$ |
| | Forgetting | $55.68 \pm 2.1$ | $60.97 \pm 1.0$ | $60.82 \pm 0.3$ | $63.46 \pm 3.9$ | $61.36 \pm 0.4$ | $56.91 \pm 1.3$ |
| | GraphCut | $62.06 \pm 1.9$ | $\mathbf{64.74 \pm 0.5}$ | $63.03 \pm 2.0$ | $61.17 \pm 1.9$ | $62.95 \pm 1.5$ | $56.91 \pm 1.3$ |
| | Random | $52.44 \pm 5.4$ | $52.34 \pm 4.3$ | $53.22 \pm 2.8$ | $53.93 \pm 4.2$ | $53.93 \pm 3.0$ | $54.79 \pm 2.9$ |
| | Herding | $32.00 \pm 2.2$ | $39.91 \pm 8.3$ | $46.91 \pm 3.3$ | $52.82 \pm 2.6$ | $51.34 \pm 1.2$ | $54.79 \pm 2.9$ |
| FOSTER (Wang et al., 2022a) | Uncertainty | $45.42 \pm 3.6$ | $49.18 \pm 4.6$ | $48.94 \pm 3.2$ | $50.95 \pm 2.6$ | $49.25 \pm 2.2$ | $54.79 \pm 2.9$ |
| | Forgetting | $45.44 \pm 3.2$ | $51.59 \pm 4.0$ | $49.37 \pm 0.2$ | $48.19 \pm 2.6$ | $49.10 \pm 1.5$ | $54.79 \pm 2.9$ |
| | GraphCut | $50.85 \pm 3.1$ | $52.54 \pm 3.7$ | $49.94 \pm 0.3$ | $49.43 \pm 0.9$ | $49.28 \pm 1.0$ | $54.79 \pm 2.9$ |
| | Random | $44.36 \pm 4.2$ | $45.41 \pm 5.5$ | $47.45 \pm 6.4$ | $48.93 \pm 7.1$ | $49.58 \pm 7.2$ | $49.22 \pm 5.5$ |
| | Herding | $39.32 \pm 0.2$ | $45.04 \pm 0.4$ | $47.90 \pm 3.1$ | $49.98 \pm 6.1$ | $49.34 \pm 6.3$ | $49.22 \pm 5.5$ |
| MEMO (Zhou et al., 2022) | Uncertainty | $38.27 \pm 6.9$ | $41.10 \pm 5.0$ | $44.99 \pm 6.4$ | $47.75 \pm 6.0$ | $47.90 \pm 5.4$ | $49.22 \pm 5.5$ |
| | Forgetting | $35.04 \pm 4.1$ | $45.23 \pm 5.4$ | $47.74 \pm 5.3$ | $48.66 \pm 5.5$ | $47.78 \pm 5.9$ | $49.22 \pm 5.5$ |
| | GraphCut | $51.37 \pm 3.6$ | $\mathbf{52.54 \pm 2.3}$ | $49.67 \pm 4.0$ | $49.97 \pm 6.0$ | $48.35 \pm 5.7$ | $49.22 \pm 5.5$ |
| | Random | $47.70 \pm 4.3$ | $55.41 \pm 5.4$ | $54.56 \pm 5.8$ | $57.75 \pm 7.5$ | $57.29 \pm 6.3$ | $59.54 \pm 8.0$ |
| | Herding | $40.32 \pm 5.0$ | $42.99 \pm 3.3$ | $54.02 \pm 4.5$ | $58.60 \pm 6.7$ | $59.11 \pm 6.3$ | $59.54 \pm 8.0$ |
| iCaRL (Rebuffi et al., 2017) | Uncertainty | $50.77 \pm 1.5$ | $54.41 \pm 6.2$ | $56.78 \pm 6.3$ | $57.38 \pm 6.6$ | $57.82 \pm 7.1$ | $59.54 \pm 8.0$ |
| | Forgetting | $53.79 \pm 4.9$ | $57.86 \pm 5.9$ | $58.30 \pm 5.9$ | $58.90 \pm 6.3$ | $56.90 \pm 7.7$ | $59.54 \pm 8.0$ |
| | GraphCut | $\mathbf{61.70 \pm 2.7}$ | $61.07 \pm 4.2$ | $60.88 \pm 5.6$ | $58.80 \pm 7.0$ | $57.68 \pm 7.1$ | $59.54 \pm 8.0$ |
| | Random | $51.02 \pm 2.7$ | $56.32 \pm 6.2$ | $57.79 \pm 4.6$ | $57.20 \pm 6.0$ | $57.77 \pm 6.9$ | $58.51 \pm 6.4$ |
| | Herding | $41.06 \pm 7.5$ | $47.97 \pm 4.0$ | $55.87 \pm 4.9$ | $58.93 \pm 4.6$ | $58.85 \pm 4.9$ | $58.51 \pm 6.4$ |
| ER (Rolnick et al., 2019) | Uncertainty | $52.70 \pm 2.4$ | $52.99 \pm 1.1$ | $56.35 \pm 6.3$ | $57.48 \pm 6.4$ | $58.09 \pm 5.4$ | $58.51 \pm 6.4$ |
| | Forgetting | $52.44 \pm 3.4$ | $55.05 \pm 5.8$ | $57.43 \pm 5.7$ | $57.00 \pm 5.5$ | $56.73 \pm 6.2$ | $58.51 \pm 6.4$ |
| | GraphCut | $\mathbf{63.03 \pm 3.1}$ | $60.53 \pm 2.6$ | $60.34 \pm 4.4$ | $58.69 \pm 5.6$ | $57.61 \pm 5.8$ | $58.51 \pm 6.4$ |
| | Random | $31.60 \pm 0.8$ | $41.46 \pm 1.9$ | $45.64 \pm 1.5$ | $51.21 \pm 4.7$ | $\mathbf{51.83 \pm 2.1}$ | $51.15 \pm 4.3$ |
| | Herding | $15.27 \pm 3.8$ | $23.75 \pm 3.0$ | $20.72 \pm 0.7$ | $27.74 \pm 5.2$ | $30.86 \pm 4.1$ | $51.15 \pm 4.3$ |
| LwF (Li & Hoiem, 2017) | Uncertainty | $26.89 \pm 5.0$ | $24.21 \pm 3.3$ | $28.95 \pm 5.1$ | $29.58 \pm 5.8$ | $30.54 \pm 4.2$ | $51.15 \pm 4.3$ |
| | Forgetting | $27.10 \pm 5.3$ | $25.49 \pm 4.0$ | $27.66 \pm 5.2$ | $30.24 \pm 5.5$ | $30.57 \pm 5.0$ | $51.15 \pm 4.3$ |
| | GraphCut | $25.34 \pm 3.1$ | $26.22 \pm 3.5$ | $29.42 \pm 5.2$ | $30.54 \pm 4.2$ | $30.95 \pm 5.4$ | $51.15 \pm 4.3$ |

# 5 Results and Analysis

In Section 5.1, we conduct a thorough analysis across diverse CIL methods and different coreset selection algorithms with varying coreset sizes. In Section 5.2, we investigate why coreset selection improves incremental accuracy, offering insight into the stability-plasticity dynamics of each CIL method. In Section 5.3, we seek to understand how these dynamics are reflected in the learning perception of the model.

## 5.1 Data diet enhances incremental performance

**Large number of samples per task.** Our analysis reveals a consistent trend of performance enhancement across various continual learners when utilizing coreset selection strategies (see Table 1). We find that when the coreset size is large enough, all selection methods tend to exhibit comparable performance. Conversely, in scenarios where the coreset size is highly reduced and restricted, a sophisticated method like GraphCut outperforms others. Moreover, the size of the coreset also plays a role: smaller coresets tend to yield more significant improvements due to increased distinction between representations which we discuss more in detail in Section 5.3. This observation is particularly evident in the case of DER which demonstrates a remarkable enhancement of approximately 7% in performance when trained only with 20% of the samples from each task.

**Small number of samples per task.** When the number of samples per task is relatively limited, we still observe performance enhancements, with mostly Uncertainty and Herding, yet they are not as pronounced due to the increased challenge of selecting informative samples (see in Table 2 and 3). Consequently, in such situations, opting for a larger coreset is more beneficial since a smaller coreset size would result in an exceptionally small sample size per task, posing a challenge for CIL. For instance, in Table 2, iCaRL improves its performance by around 3% when trained with 80% of the samples from each task, compared to full sample training. However, its performance stars to degrade when coreset size is less than 50%.

Table 2: Accuracy [%] of CIL models across various coreset fractions and selections on **Split-CIFAR100**. The best results are highlighted in bold if coreset selection outperforms training with all samples.

| | Fraction | 10% | 20% | 50% | 80% | 90% | 100% |
|---|---|---|---|---|---|---|---|
| | Random | $26.23 \pm 0.6$ | $36.35 \pm 2.8$ | $47.32 \pm 2.6$ | $53.11 \pm 1.6$ | $54.07 \pm 0.1$ | $53.81 \pm 1.0$ |
| | Herding | $17.99 \pm 7.5$ | $24.79 \pm 6.0$ | $41.11 \pm 2.7$ | $52.48 \pm 0.4$ | $53.92 \pm 0.8$ | $53.81 \pm 1.0$ |
| DER (Yan et al., 2021) | Uncertainty | $27.54 \pm 4.6$ | $38.29 \pm 3.0$ | $49.41 \pm 1.2$ | $\mathbf{55.71 \pm 1.9}$ | $54.55 \pm 0.4$ | $53.81 \pm 1.0$ |
| | Forgetting | $30.32 \pm 4.9$ | $41.25 \pm 1.8$ | $49.20 \pm 2.2$ | $54.10 \pm 0.3$ | $53.68 \pm 0.1$ | $53.81 \pm 1.0$ |
| | GraphCut | $29.61 \pm 5.7$ | $39.71 \pm 3.4$ | $50.35 \pm 1.0$ | $53.08 \pm 0.8$ | $54.89 \pm 0.7$ | $53.81 \pm 1.0$ |
| | Random | $23.21 \pm 0.0$ | $32.04 \pm 1.3$ | $48.95 \pm 0.8$ | $51.71 \pm 1.9$ | $53.34 \pm 0.8$ | $56.19 \pm 2.3$ |
| | Herding | $10.84 \pm 0.8$ | $18.38 \pm 1.1$ | $35.15 \pm 2.7$ | $51.51 \pm 0.1$ | $53.72 \pm 0.9$ | $56.19 \pm 2.3$ |
| FOSTER (Wang et al., 2022a) | Uncertainty | $16.97 \pm 0.1$ | $27.37 \pm 0.9$ | $44.29 \pm 3.1$ | $55.24 \pm 0.1$ | $55.10 \pm 1.7$ | $56.19 \pm 2.3$ |
| | Forgetting | $21.80 \pm 0.4$ | $32.42 \pm 0.8$ | $44.97 \pm 2.9$ | $54.59 \pm 0.4$ | $54.91 \pm 1.0$ | $56.19 \pm 2.3$ |
| | GraphCut | $22.16 \pm 1.6$ | $30.40 \pm 1.1$ | $45.91 \pm 2.3$ | $53.35 \pm 1.9$ | $55.24 \pm 0.5$ | $56.19 \pm 2.3$ |
| | Random | $20.79 \pm 0.7$ | $26.74 \pm 0.1$ | $29.62 \pm 0.5$ | $34.58 \pm 0.1$ | $34.58 \pm 0.1$ | $34.23 \pm 0.4$ |
| | Herding | $13.24 \pm 2.0$ | $18.76 \pm 1.5$ | $27.26 \pm 1.8$ | $33.64 \pm 0.3$ | $\mathbf{34.94 \pm 0.1}$ | $34.23 \pm 0.4$ |
| MEMO (Zhou et al., 2022) | Uncertainty | $16.07 \pm 2.6$ | $23.23 \pm 2.9$ | $30.14 \pm 1.7$ | $33.41 \pm 0.9$ | $34.10 \pm 1.0$ | $34.23 \pm 0.4$ |
| | Forgetting | $18.44 \pm 1.9$ | $23.37 \pm 2.0$ | $31.17 \pm 0.3$ | $33.10 \pm 0.4$ | $32.46 \pm 2.2$ | $34.23 \pm 0.4$ |
| | GraphCut | $23.21 \pm 1.7$ | $27.79 \pm 0.6$ | $32.49 \pm 0.6$ | $33.61 \pm 0.2$ | $34.22 \pm 0.7$ | $34.23 \pm 0.4$ |
| | Random | $25.48 \pm 0.2$ | $29.87 \pm 3.0$ | $35.37 \pm 2.0$ | $37.02 \pm 3.1$ | $37.11 \pm 3.0$ | $37.45 \pm 1.7$ |
| | Herding | $13.02 \pm 1.2$ | $17.24 \pm 1.5$ | $27.91 \pm 1.3$ | $38.24 \pm 1.3$ | $37.55 \pm 0.8$ | $37.45 \pm 1.7$ |
| iCaRL (Rebuffi et al., 2017) | Uncertainty | $22.47 \pm 1.9$ | $28.05 \pm 1.3$ | $35.18 \pm 3.3$ | $\mathbf{40.25 \pm 0.7}$ | $39.26 \pm 2.5$ | $37.45 \pm 1.7$ |
| | Forgetting | $25.00 \pm 0.3$ | $27.80 \pm 1.1$ | $33.27 \pm 2.0$ | $37.80 \pm 1.0$ | $37.44 \pm 2.2$ | $37.45 \pm 1.7$ |
| | GraphCut | $24.04 \pm 0.7$ | $30.45 \pm 0.2$ | $33.31 \pm 0.3$ | $35.76 \pm 3.2$ | $38.03 \pm 0.8$ | $37.45 \pm 1.7$ |
| | Random | $25.23 \pm 0.3$ | $31.58 \pm 3.0$ | $37.64 \pm 1.4$ | $39.25 \pm 1.3$ | $40.66 \pm 2.0$ | $39.53 \pm 1.6$ |
| | Herding | $19.13 \pm 5.4$ | $24.90 \pm 6.3$ | $34.92 \pm 4.0$ | $40.18 \pm 2.1$ | $\mathbf{41.19 \pm 1.2}$ | $39.53 \pm 1.6$ |
| ER (Rolnick et al., 2019) | Uncertainty | $25.77 \pm 4.6$ | $31.63 \pm 4.3$ | $36.61 \pm 1.5$ | $41.14 \pm 0.4$ | $39.69 \pm 1.4$ | $39.53 \pm 1.6$ |
| | Forgetting | $29.53 \pm 4.7$ | $33.97 \pm 3.8$ | $36.96 \pm 3.4$ | $40.58 \pm 0.7$ | $39.92 \pm 2.5$ | $39.53 \pm 1.6$ |
| | GraphCut | $32.99 \pm 8.7$ | $38.22 \pm 6.4$ | $39.55 \pm 3.5$ | $39.61 \pm 2.6$ | $39.97 \pm 0.6$ | $39.53 \pm 1.6$ |
| | Random | $11.39 \pm 1.0$ | $15.38 \pm 1.3$ | $20.26 \pm 1.3$ | $22.93 \pm 2.1$ | $\mathbf{23.91 \pm 1.2}$ | $22.82 \pm 1.4$ |
| | Herding | $3.67 \pm 1.3$ | $6.22 \pm 0.1$ | $12.43 \pm 2.0$ | $17.09 \pm 4.6$ | $18.08 \pm 4.5$ | $22.82 \pm 1.4$ |
| LwF (Li & Hoiem, 2017) | Uncertainty | $9.55 \pm 0.5$ | $12.17 \pm 1.8$ | $15.54 \pm 2.8$ | $18.72 \pm 5.0$ | $18.00 \pm 4.2$ | $22.82 \pm 1.4$ |
| | Forgetting | $9.93 \pm 1.3$ | $12.75 \pm 2.7$ | $15.18 \pm 2.9$ | $17.99 \pm 4.5$ | $18.28 \pm 4.4$ | $22.82 \pm 1.4$ |
| | GraphCut | $8.17 \pm 0.3$ | $10.37 \pm 1.4$ | $15.56 \pm 3.4$ | $17.26 \pm 4.1$ | $18.00 \pm 4.9$ | $22.82 \pm 1.4$ |

**Experiments on pretrained backbone.** We further complemented our study with pretrained ResNet18 and ViT backbones where the results align with the findings discussed herein. We observe that pretraining improves the performance regardless of coreset selection. However, coreset selection provides an additional performance boost. For more details, specifically for CODA-Prompt, please refer to the Appendix A.5.

**FOSTER benefits from more samples.** FOSTER's primary objective is to identify critical elements that were potentially overlooked or misinterpreted by the original model during the learning process. For instance, in the initial stages of learning, certain features may have been deemed less significant than others. However, as the model progresses and encounters new concepts, previously redundant features may become crucial. FOSTER addresses these dynamics by employing a feature-boosting mechanism, which aims to highlight the evolving importance of features over time. However, this mechanism may necessitate access to more samples to effectively capture the intricate relationships between features. Consequently, training with the full dataset enables the model to develop a more comprehensive understanding of the underlying patterns and correlations among the features.

**LwF exhibits abrupt weight changes when trained with a coreset.** Sophisticated coreset selection approaches do not yield performance advantages in LwF. Surprisingly, learning from a random samples appears to drive improvements instead. To understand this phenomenon, we conduct an in-depth investigation, focusing on the performance after each task, as illustrated in Figure 2. Our analysis shows that LwF trained with more advanced coreset selection methods, such as Uncertainty and GraphCut, demonstrate superior adaptability to the current task. However, this enhanced adaptability comes at a cost of catastrophic forgetting. To unravel the root cause of this forgetting phenomenon, we examine the changes in model parameters between consecutive tasks. We found that Uncertainty and GraphCut induce abrupt changes in the parameters, whereas it is comparatively smaller with randomly selected samples.

Table 3: Accuracy [%] of CIL models across various coreset fractions and selections on **Split-ImageNet100**. Learning from coreset samples enhances the performance, except FOSTER and LwF. The best results are highlighted in bold if coreset outperforms training with all samples.

| | Fraction | 10% | 20% | 50% | 80% | 90% | 100% |
|---|---|---|---|---|---|---|---|
| | Random | $19.89 \pm 2.3$ | $32.70 \pm 1.5$ | $42.45 \pm 0.6$ | $52.61 \pm 1.8$ | $53.12 \pm 1.0$ | $55.03 \pm 1.2$ |
| | Herding | $18.30 \pm 1.2$ | $29.83 \pm 0.6$ | $44.77 \pm 0.8$ | $53.59 \pm 0.3$ | $55.52 \pm 0.1$ | $55.03 \pm 1.2$ |
| DER (Yan et al., 2021) | Uncertainty | $27.08 \pm 0.5$ | $36.92 \pm 0.9$ | $49.84 \pm 0.4$ | $55.10 \pm 0.2$ | $\mathbf{56.46 \pm 0.6}$ | $55.03 \pm 1.2$ |
| | Forgetting | $32.69 \pm 2.1$ | $40.21 \pm 1.3$ | $50.27 \pm 0.9$ | $55.15 \pm 0.7$ | $55.60 \pm 0.8$ | $55.03 \pm 1.2$ |
| | GraphCut | $32.91 \pm 0.7$ | $38.90 \pm 0.4$ | $50.12 \pm 0.8$ | $54.71 \pm 0.3$ | $55.81 \pm 0.1$ | $55.03 \pm 1.2$ |
| | Random | $17.59 \pm 1.3$ | $22.68 \pm 0.8$ | $34.20 \pm 3.8$ | $46.90 \pm 4.1$ | $48.64 \pm 4.2$ | $52.06 \pm 0.4$ |
| | Herding | $8.67 \pm 0.1$ | $13.42 \pm 0.2$ | $30.63 \pm 1.7$ | $45.85 \pm 1.0$ | $48.89 \pm 0.1$ | $52.06 \pm 0.4$ |
| FOSTER (Wang et al., 2022a) | Uncertainty | $8.14 \pm 0.1$ | $15.91 \pm 0.5$ | $35.40 \pm 0.5$ | $46.39 \pm 0.5$ | $48.37 \pm 0.5$ | $52.06 \pm 0.4$ |
| | Forgetting | $11.62 \pm 0.5$ | $18.71 \pm 0.4$ | $35.26 \pm 0.3$ | $46.95 \pm 0.9$ | $49.45 \pm 0.4$ | $52.06 \pm 0.4$ |
| | GraphCut | $16.74 \pm 0.5$ | $22.99 \pm 0.1$ | $37.42 \pm 0.4$ | $47.22 \pm 0.4$ | $49.95 \pm 0.9$ | $52.06 \pm 0.4$ |
| | Random | $18.79 \pm 0.1$ | $27.29 \pm 0.2$ | $40.02 \pm 1.7$ | $44.48 \pm 0.2$ | $47.80 \pm 1.9$ | $46.36 \pm 1.0$ |
| | Herding | $18.15 \pm 1.1$ | $26.08 \pm 0.4$ | $37.71 \pm 3.1$ | $46.76 \pm 2.3$ | $47.94 \pm 1.1$ | $46.36 \pm 1.0$ |
| MEMO (Zhou et al., 2022) | Uncertainty | $20.22 \pm 0.8$ | $26.94 \pm 2.2$ | $39.39 \pm 1.1$ | $45.90 \pm 0.4$ | $\mathbf{48.54 \pm 0.2}$ | $46.36 \pm 1.0$ |
| | Forgetting | $24.40 \pm 1.5$ | $33.16 \pm 1.0$ | $41.86 \pm 0.5$ | $45.57 \pm 0.5$ | $47.19 \pm 0.9$ | $46.36 \pm 1.0$ |
| | GraphCut | $29.76 \pm 1.8$ | $35.73 \pm 1.1$ | $42.80 \pm 1.9$ | $45.98 \pm 2.8$ | $48.50 \pm 1.3$ | $46.36 \pm 1.0$ |
| | Random | $21.93 \pm 0.7$ | $27.29 \pm 0.5$ | $30.21 \pm 3.7$ | $29.12 \pm 1.9$ | $30.30 \pm 1.6$ | $33.05 \pm 1.8$ |
| | Herding | $20.80 \pm 1.8$ | $24.29 \pm 2.3$ | $30.92 \pm 0.2$ | $33.23 \pm 0.9$ | $34.04 \pm 0.2$ | $33.05 \pm 1.8$ |
| iCaRL (Rebuffi et al., 2017) | Uncertainty | $22.52 \pm 0.3$ | $22.37 \pm 0.9$ | $32.67 \pm 1.6$ | $33.03 \pm 0.1$ | $34.76 \pm 0.9$ | $33.05 \pm 1.8$ |
| | Forgetting | $26.38 \pm 0.1$ | $28.35 \pm 0.8$ | $31.85 \pm 0.7$ | $33.80 \pm 0.6$ | $34.77 \pm 2.7$ | $33.05 \pm 1.8$ |
| | GraphCut | $33.04 \pm 0.6$ | $35.10 \pm 0.6$ | $34.87 \pm 1.1$ | $\mathbf{35.19 \pm 0.2}$ | $31.29 \pm 0.3$ | $33.05 \pm 1.8$ |
| | Random | $20.19 \pm 0.1$ | $25.84 \pm 2.7$ | $30.47 \pm 2.0$ | $29.14 \pm 1.0$ | $30.81 \pm 0.6$ | $34.23 \pm 4.2$ |
| | Herding | $20.21 \pm 0.1$ | $24.56 \pm 0.8$ | $29.81 \pm 1.1$ | $31.92 \pm 0.4$ | $33.68 \pm 0.7$ | $34.23 \pm 4.2$ |
| ER (Rolnick et al., 2019) | Uncertainty | $20.82 \pm 0.6$ | $23.08 \pm 0.6$ | $29.23 \pm 0.5$ | $29.35 \pm 1.1$ | $30.74 \pm 1.4$ | $34.23 \pm 4.2$ |
| | Forgetting | $24.85 \pm 0.6$ | $28.32 \pm 1.4$ | $29.03 \pm 0.2$ | $32.85 \pm 0.4$ | $31.74 \pm 2.1$ | $34.23 \pm 4.2$ |
| | GraphCut | $30.13 \pm 1.0$ | $30.52 \pm 0.2$ | $\mathbf{34.83 \pm 0.6}$ | $32.05 \pm 1.5$ | $32.16 \pm 0.5$ | $34.23 \pm 4.2$ |
| | Random | $9.25 \pm 0.1$ | $11.22 \pm 0.7$ | $15.88 \pm 0.8$ | $16.27 \pm 1.1$ | $\mathbf{16.52 \pm 0.5}$ | $16.46 \pm 1.8$ |
| | Herding | $5.70 \pm 0.5$ | $7.65 \pm 1.1$ | $10.70 \pm 0.1$ | $11.33 \pm 0.2$ | $11.64 \pm 0.2$ | $16.46 \pm 1.8$ |
| LwF (Li & Hoiem, 2017) | Uncertainty | $7.84 \pm 0.1$ | $8.07 \pm 0.1$ | $11.27 \pm 0.2$ | $11.41 \pm 0.1$ | $11.51 \pm 0.3$ | $16.46 \pm 1.8$ |
| | Forgetting | $7.38 \pm 0.2$ | $10.01 \pm 0.1$ | $11.60 \pm 0.2$ | $12.15 \pm 0.1$ | $12.57 \pm 0.3$ | $16.46 \pm 1.8$ |
| | GraphCut | $7.41 \pm 0.2$ | $9.29 \pm 0.8$ | $10.77 \pm 0.5$ | $12.06 \pm 0.2$ | $12.88 \pm 0.1$ | $16.46 \pm 1.8$ |

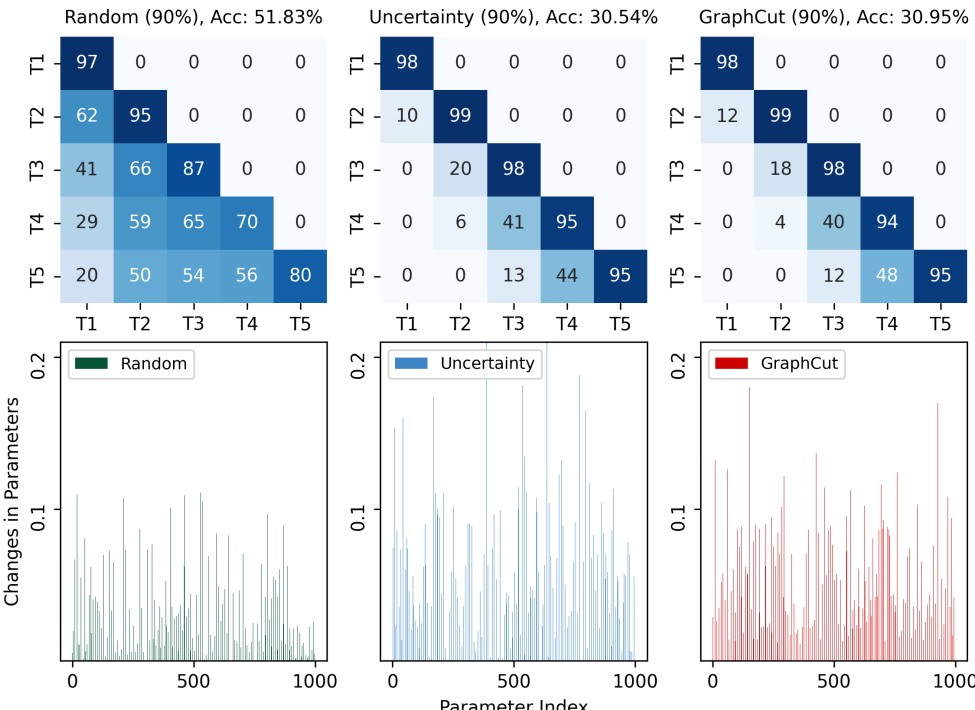

Figure 2: Accuracy [%] on Split-CIFAR10 after each learning step on LwF *(above)*, reveals that Random selection demonstrates relatively less forgetting while effectively learning. This is due to the abrupt parameter changes. For example, on the last layer between consecutive tasks *(below)*, Uncertainty and GraphCut abruptly shift the parameters.

This is because, coreset selection strategies (e.g., herding, uncertainty, and graph cut) prioritize the most informative samples specific to the current task. When the CL approach relies solely on regularization, this prioritization can lead to overfitting to the current task's distribution. Such overfitting amplifies significant representation shifts, resulting in abrupt parameter updates that results in catastrophic forgetting of previously learned tasks. In contrast, random selection is implicitly incorporates as a form of regularizer with a greater diversity and variability in the sample distribution across tasks, which helps to mitigate overfitting and results in more stable parameter updates. This suggests that the traditional regularization methods may not be as effective as replay-based approaches when considering coreset utilization.

## 5.2 Incremental performance increases because models forget less

The performance improvements observed in class-incremental learners when trained on coreset samples can be attributed to several factors:

(i) First, coreset samples are carefully selected to represent the most informative subset of the data, thereby reducing redundancy and focusing on critical information. This strategy enhances the model's capacity for retention of essential information while minimizing the risk of overfitting to less relevant data points. In other words, this allows for more focused exposure to relevant data and develops robust representations that consolidate the acquired knowledge better, leading to improved performance in most class-incremental learning baselines in our analysis.

(ii) Second, sample selection before training is also crucial in enhancing the data quality utilized during the replay or memory construction phase in continual learning. By filtering out potentially irrelevant or redundant data points beforehand, it ensures that only the most informative and representative samples are stored in memory. This contributes to enhanced retention or consolidation of learned knowledge from previous tasks over time by focusing on key patterns and relationships.

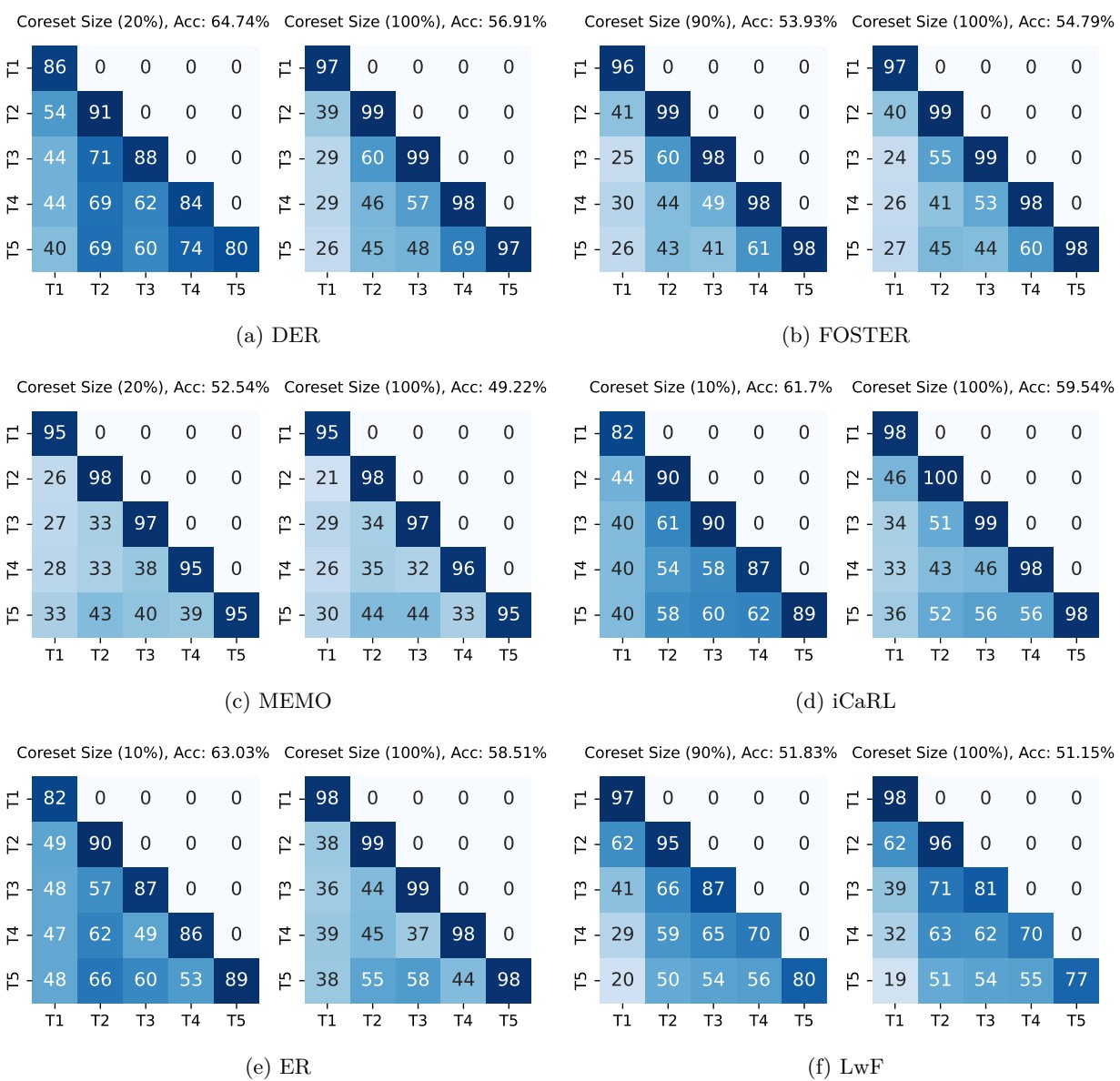

Figure 3: Accuracy [%] of each task after every learning session on different class-incremental learning methods with Split-CIFAR10. This comparison includes the performance using all samples *vs.* the best performing coreset selection, which may involve different coreset fractions. The underlying reason for the improved accuracy is attributed to reduced forgetting.

Consequently; DER, iCaRL, and ER demonstrate noticeable improvement in knowledge retention learning when trained on coreset samples (see Figure 3). These methods leverage the enhanced representativeness and diversity of coreset samples, reinforcing old knowledge retention while learning new ones. MEMO and LwF also benefit from training on coreset samples, albeit to a lesser extent. FOSTER still appears to rely more heavily on learning from the complete dataset, maintaining consistent performance across tasks. This reaffirms that its learning strategy may be better suited to leveraging the full dataset rather than coreset samples as we discuss above. In the Appendix A.4, we also provide more details and share the accuracy per task after each learning session on Split-CIFAR100. Overall, our analysis indicates that the enhanced incremental performance with coreset selection is primarily attributed to knowledge retention.

## 5.3    Models that forget less, preserve the representations better

Here, we delve deeper into the key factor that drives enhanced knowledge retention. Specifically, we aim to explore how different class-incremental learners' perceptions evolved under different coreset methods and fractions. To achieve this, we generate saliency maps, as illustrated in Figure 4, with the objective of discerning where the model directed its attention after being trained with a coreset and compare against all data samples. We find that models trained with the coresets exhibit a greater ability to retain focus on the object itself, effectively capturing the essence of the image. In contrast, models trained on all data samples tend to shift their focus to areas outside the main object. This insight sheds light on our earlier discussion regarding the model's knowledge retention or *not* forgetting ability, and highlights that coreset selection gives more attention to relevant features.

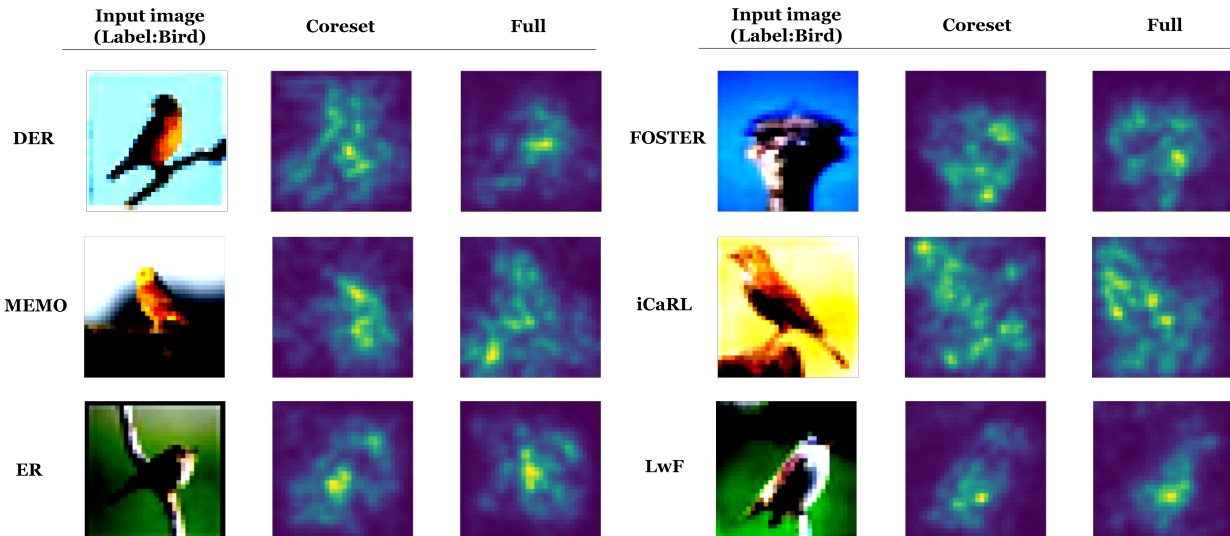

Figure 4: Saliency maps from the first encountered task after completing all learning sessions. Models trained with selected coresets exhibit enhanced perception capabilities in capturing the important parts of an input. Note that we select top performing coreset selection methods across different class-incremental learners.

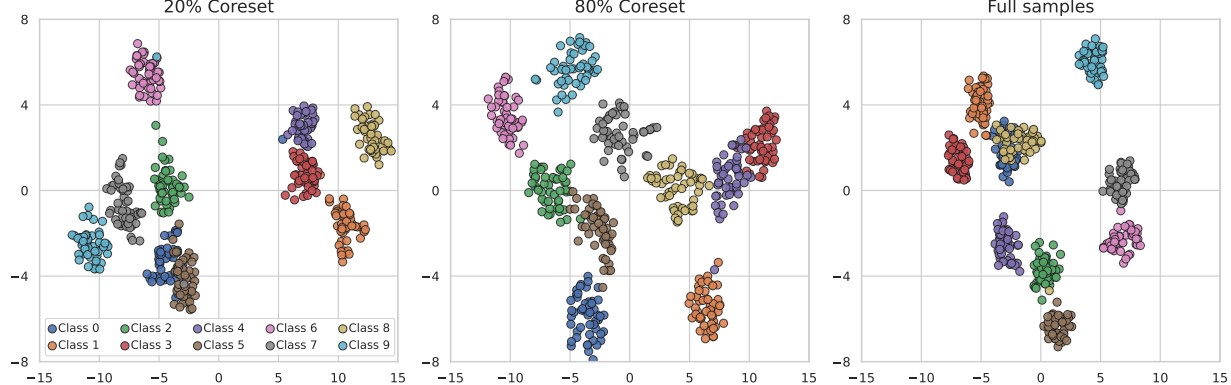

Figure 5: DER's representation of all classes on Split-CIFAR10 with varying coresets selected with GraphCut, compared to the full samples. When it is trained with coresets, it exhibits superior ability to distinct representations.

Furthermore, we investigate how the model's representation ability evolves as the coreset size changes, providing insights on the relationship between coreset composition and class separability. To illustrate this, in Figure 5, we employ DER to examine its representation of each class after completing all learning sessions. Notably, when using a smaller coreset, such as 20%, the model demonstrates distinct separations between classes, effectively preserving boundaries between different categories. This suggests that with fewer, more concentrated samples, the model can maintain clearer distinctions.

However, as the coreset size increases, we observe a noticeable convergence in class representations, with boundaries between classes becoming less distinct. This trend suggests that larger coresets, while offering more data, may introduce redundancy or noise, causing overlap between classes and ultimately increasing the misclassification during inference. This phenomenon underscores the delicate balance between data quantity and quality, where more data does not necessarily translate into better generalization in class-incremental learning.

# 6 Conclusion

Existing CL approaches predominantly use all available data during training yet not all samples carry equal informational value and not need to go under the training process. In this study, we explore the underutilized potential of selective learning from key samples, demonstrating that model performance is strongly influenced by both the quality and quantity of data. Our empirical analysis yields three key findings that challenge and extend current CL methodologies. First, we show that learning from coreset samples enhances incremental performance. We attribute this improvement to better knowledge retention across tasks, achieved by reducing redundancy and focusing on high-value information. Further, we observe that models trained with coresets exhibit a refined perception, capturing essential features of input data more effectively and maintaining clearer class distinctions by the end of all sessions. Additionally, our results reveal that the effectiveness of coreset selection algorithms is highly context dependent, varying with both the chosen method and the specific dataset. This underscores the necessity for CL-tailored coreset strategies to optimize performance across diverse scenarios. These findings underscore the substantial impact of learning from coreset samples on continual learning, and aims to provide a foundation for designing more effective CL models for practical applications. Future studies could extend this work by examining coreset strategies in online or blurry class-incremental learning contexts, potentially enhancing adaptability and efficiency in real-world scenarios.

## Broader Impact Statement

This paper aims to advance the field of Machine Learning, especially on the subject of Class-Incremental Learning. Besides the advancements in the field, it shows training with smaller but more representative samples improves performance, thereby reducing memory and computation concerns.

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

# A  Appendix

In this appendix, we first give more detailed explanations of the continual learning methods and coreset selection methods used in our experiments. This includes a comprehensive overview of the baseline methods and their key characteristics. Next, we provide more details about our implementation for the backbones we used and the metrics that we evaluated. Then, we share the accuracy of each task after every learning session for the Split-CIFAR100 dataset trained with ResNet18, similar to Figure 3. Finally, we provide more results with pretrained ResNet18 and pretrained ViT on Split-CIFAR10 and Split-CIFAR100.

## A.1  Continual Learning Approaches

In our evaluation, we selected a diverse set of continual learning methods to ensure a comprehensive analysis, including regularization-based, replay-based, architecture-based, and prompt-based approaches. In total, we evaluated seven different methods.

### A.1.1  Regularization-based Methods

Regularization-based methods utilize a single backbone, meaning they rely on one fixed architecture without altering its structure. These methods operate without accessing any memory data, working solely with the data from the current task. This constraint makes them particularly challenging compared to other approaches. To mitigate catastrophic forgetting, these methods regularize weight updates during the learning of each new task. By carefully controlling the extent of weight changes, they ensure that the model retains knowledge from previous tasks.

**LwF** is one of the most well-known and well-established regularization approaches in continual learning. It tackles catastrophic forgetting by leveraging knowledge distillation to transfer knowledge from a previously trained model (the teacher) to the current model (the student) as new tasks are introduced. When training on a new task, LwF preserves the knowledge of earlier tasks by ensuring the current model reproduces the predictions of the teacher model for the classes associated with prior tasks. Specifically, the teacher model is frozen after completing a task and generates *soft labels* for the new training data, which represent the probability distribution over previously learned classes. Formally, the learning process is guided by two losses: cross-entropy loss $\mathcal{L}_{\mathrm{CE}}$ given in Eq 2 where $y_i$ is the true label and $p_i$ is the predicted probability for the $i$-th input, and the distillation loss $\mathcal{L}_{\mathrm{KL}}$ given in Eq 3 where $q_{\mathrm{teacher}}(x_i)$ is the probability distribution from the teacher model and $q_{\mathrm{student}}(x_i)$ is the probability distribution from the current model for the same input.

$$L_{\mathrm{CE}} = -\sum_{i=1}^{N} y_i \log(p_i) \tag{2}$$

$$L_{\mathrm{KL}} = \sum_{i=1}^{N} \mathrm{KL}\left(q_{\mathrm{teacher}}(x_i) \parallel q_{\mathrm{student}}(x_i)\right) \tag{3}$$

Cross-entropy loss ensures that the model performs well on the current task and the distillation loss helps the model retain knowledge from previously learned tasks. It measures the difference between the predicted probability distributions of the current model and the teacher model for previously seen examples. This is typically calculated using the Kullback-Leibler (KL) divergence.

Finally, the CL loss $\mathcal{L}_{\mathrm{CL}}$ for LwF is the combination of the cross-entropy loss $\mathcal{L}_{\mathrm{CE}}$ and the distillation loss $\mathcal{L}_{\mathrm{KL}}$ with a scaling factor of $\lambda$ that controls the importance of the distillation loss:

$$L_{\mathrm{CL}}^{\mathrm{LwF}} = L_{\mathrm{CE}} + \lambda L_{\mathrm{KL}} \tag{4}$$

### A.1.2   Replay-based Methods

Replay-based methods, on the other hand, employ an additional memory buffer to store a subset of past task data. While learning new tasks, these methods simultaneously utilize the memory buffer samples $M$ together with current task samples $N$, allowing the model to retain a degree of knowledge about previous tasks. This mechanism provides a practical way to alleviate forgetting.

**ER** is a key replay-based method in continual learning. It maintains a memory buffer containing data from previous tasks and combines this replayed data with the new task data. The model then computes the cross-entropy loss, given in Eq. 2, to evaluate how well the model's predictions align with the true labels for both the current task and the replayed task samples.

$$L_{\mathrm{CL}}^{\mathrm{ER}} = -\sum_{i=1}^{N+M} y_i \log(p_i) \tag{5}$$

**iCaRL** differs from the ER method by introducing a specific memory selection strategy, known as herding, and incorporating a distillation loss into its training objective. While ER relies solely on cross-entropy loss for current and replayed data, iCaRL combines cross-entropy loss Eq. 2 with a distillation loss Eq. 3 to strengthen its knowledge retention mechanism, similar to LwF.

$$L_{\mathrm{CL}}^{\mathrm{iCaRL}} = -\sum_{i=1}^{N+M} y_i \log(p_i) + \lambda L_{\mathrm{KL}} \tag{6}$$

### A.1.3   Architecture-based Methods

Architecture-based methods take a different approach by dynamically modifying the model's backbone. When encountering a new task, these methods either create a completely new architecture (a new model) or initialize additional components. The newly added parts are then trained specifically on the new task data, enabling the model to adapt structurally to task-specific requirements and reinforce knowledge retention.

**DER** initializes a new backbone for each task and aggregates features from both old (frozen) and new backbones using an expanded fully connected layer. This enables the model to specialize for new tasks while preserving knowledge from earlier tasks. A key component of DER is the auxiliary loss $L_{\mathrm{aux}}$ in Eq. 7 which promotes learning diverse and discriminative features for each task. Therefore, it uses temporary auxiliary classes $y^{aux}$ and classifier by treating all old classes as one category and the new classes as another. Therefore, the complete loss function for DER can be expressed as in Eq. 8 .

$$L_{\mathrm{aux}} = -\sum_{i=1}^{N+M} y_i^{aux} \log(p_i^{aux}) \tag{7}$$

$$L_{\mathrm{CL}}^{\mathrm{DER}} = -\sum_{i=1}^{N+M} y_i \log(p_i) + L_{\mathrm{aux}} \tag{8}$$

**MEMO** operates on the assumption that shallow network layers capture general patterns, while deeper layers specialize in task-specific concepts. To accommodate new tasks, MEMO initializes fresh deep layers or blocks for each task while preserving the shallow layers unchanged. Consequently, the model expands only the deep layers for new tasks. MEMO employs the same loss function as the DER method, enhanced by the inclusion of an additional lambda hyperparameter that controls the auxiliary loss:

$$L_{\text{CL}}^{\text{MEMO}} = - \sum_{i=1}^{N+M} y_i \log(p_i) - \lambda L_{\text{aux}} \tag{9}$$

**FOSTER** combines feature boosting and feature compression in two stages to alleviate forgetting. In the boosting stage, FOSTER adds a new feature extractor to the model when a new task arrives. The new feature extractor learns residual features, which capture the differences (residuals) between the target outputs and the predictions from the frozen old model. These residual features are then concatenated with the frozen old features, creating a combined representation.

In the boosting stage FOSTER benefits from 2 different classifiers. The first one maintains the balance between old and new classes by aligning the logits (Eq.10), and the second one explicitly improves the representation of old classes by using only the new feature extractor's output over all classes (Eq. 11). Finally, similar to LwF and iCaRL, knowledge distillation is applied during this stage to align the outputs of the new model with the frozen old model to further preserve the knowledge from previous tasks. Then the total loss for FOSTER can be expressed as in Eq. 12.

$$L_{\text{LA}} = - \sum_{i=1}^{N+M} y_i \log(p_i^{\text{aligned}}) \tag{10}$$

$$L_{\text{FE}} = - \sum_{i=1}^{N+M} y_i \log(p_i^{\text{enhanced}}) \tag{11}$$

$$L_{\text{CL}}^{\text{FOSTER}} = L_{\text{LA}} + L_{\text{FE}} + \lambda L_{\text{KL}} \tag{12}$$

Following the boosting stage, where a new feature extractor is added to handle residual features for new tasks, the compression process starts to address the problem of parameter growth caused by the dynamic expansion of the model. In this final stage, the dual-branch architecture (frozen old model + new feature extractor) is compressed into a single compact backbone.

### A.1.4 Prompt-based Methods

Prompt-based methods represent a recently developed approach in the field of continual learning. Drawing inspiration from prompt-tuning techniques in natural language processing, these methods use task-specific prompts to guide the model's behavior. Unlike traditional approaches that modify weights or architectures, prompt-based methods largely retain the shared backbone architecture and instead focus on learning small, task-specific prompts.

**CODA-Prompt** learns prompt components that are dynamically combined with input conditioned weights to create task-specific prompts. When a new task is introduced, a distinct prompt is initialized to capture task-specific information. This prompt interacts with the shared backbone model to activate the relevant representations for the current task. By localizing task-specific adaptations within the prompts, the model can effectively generalize across tasks while minimizing interference. This leads to the model's ability to retain previously acquired knowledge, as the large pretrained backbone remains unchanged, while still efficiently learning new tasks. Formally, to achieve this, CODA-Prompt uses the prompt loss in Eq. 13 that aims to maximize the alignment between the prompts and the task-specific features while minimizing redundancy or uninformative contributions. In this formulation, $P_i$ represents the prompt embedding for block $i$, and $Q_i$ denotes the corresponding input feature embedding for block $i$. The term $\|\cdot\|_2^2$ is the squared $L_2$-norm, which measures the difference between the prompt and input embeddings. $N$ is the number of transformer blocks with prompts. This enforces that the prompt embeddings $P_i$ are closely aligned with the input embeddings $Q_i$, ensuring task relevance.

$$L_{\text{prompt}} = \frac{1}{N} \sum_{i=1}^{N} \|P_i - Q_i\|_2^2 \tag{13}$$

Finally, CODA-Prompt combines the standard cross-entropy loss for classification with the prompt loss summed over all transformer blocks to learn the task-specific prompts on top of pretrained ViTs:

$$L_{\text{CL}}^{\text{CODA}} = -\sum_{i=1}^{N} y_i \log(p_i) + L_{\text{prompt}} \tag{14}$$

### A.2   Coreset Selection Approaches

Coreset selection refers to the process of selecting a small, representative subset of data points from a given larger original dataset $D = \{(x_i, y_i)\}_{i=1}^{N}$ where $x_i$ are the input features and $y_i$, such that the selected subset (the coreset) can approximate the performance of the full dataset for a given machine learning task.

**Random** selection is a straightforward approach to dataset reduction. In this method, a fixed number or proportion of data points is chosen uniformly at random from the original dataset. While this technique does not account for the importance or representativeness of individual samples, it serves as a strong and computationally efficient baseline. This method works by choosing $I$ samples uniformly at random without replacement from the original dataset and the selected coreset $C$ can be written as:

$$C = \{(x_i, y_i) \mid i \in I\} \tag{15}$$

**Herding** is a deterministic method for selecting a representative subset of data points, known as a coreset. It focuses on capturing the overall structure of the dataset by ensuring that the selected samples approximate the mean feature representation of the full dataset.

The method computes the mean of the features $m = \frac{1}{N} \sum_{i=1}^{N} \phi(x_i)$ where $\phi(x_i)$ maps each data point to a feature space, such as one generated by a neural network. The goal is to iteratively build a coreset $C$ that closely approximates this mean.

The algorithm starts with an empty coreset and a residual vector $r = m$, which keeps track of the difference between the dataset mean and the cumulative contributions of the selected samples. At each step, the next sample to include in the coreset is chosen by finding the data point $x_i$ whose feature vector $\phi(x_i)$ has the largest alignment with the residual vector $r$. This can be expressed as:

$$i^* = \arg\max_{i} r^\top \phi(x_i). \tag{16}$$

Once $x_{i*}$ is selected, it is added to the coreset, and the residual vector is updated by subtracting $\phi(x_{i*})$. This process is repeated until the desired number of samples is selected, resulting in the coreset $C$. By iteratively reducing the residual, herding ensures that the selected coreset is highly representative of the original dataset. This makes it a valuable approach for tasks requiring a compact yet informative subset of data.

**Uncertainty** coreset selection is a method that prioritizes data points where the model exhibits the highest uncertainty in its predictions. The rationale is that these uncertain samples carry the most informative value, as they highlight areas where the model is less confident and likely to benefit from further training.

For a model $f(x)$ that outputs a probability distribution over classes, the uncertainty of a sample $x_i$ can be quantified using measures such as entropy. The entropy for a prediction is computed as $H(x_i) = -\sum_{c \in C} p_c(x_i) \log p_c(x_i)$ where $p_c(x_i)$ is the predicted probability for class $c$, and $C$ is the set of all possible classes. The uncertainty selection process involves computing $H(x_i)$ for all samples in the dataset and ranking them by their uncertainty scores. The top $k$ samples with the highest entropy are chosen to form the coreset:

$$C = \{(x_i, y_i) \mid x_i \text{ ranks among the top } k \text{ in } H(x_i)\}. \tag{17}$$

By selecting the most uncertain samples, this method focuses on the regions of the data space where the model requires additional learning, ensuring an informative and compact coreset, particularly effective when resources for training are limited.

**Forgetting** coreset selection identifies data points that the model struggles to consistently classify correctly during training. These are known as "forgotten examples" because their predictions frequently change from correct to incorrect. By focusing on such challenging samples, this method selects a subset of data that is highly informative for improving the model's robustness.

During training, the model keeps track of whether it correctly predicts each sample at every training step. Let's denote the prediction correctness for a sample $x_i$ at a given step as a binary value; 1 if the prediction is correct and 0 if the prediction is incorrect.

A *forgetting event* occurs when the model's prediction for a sample changes from correct to incorrect as training progresses. The forgetting score for a sample is simply the total number of forgetting events it experiences during training. For example, if the prediction for $x_i$ flips from correct to incorrect three times, its forgetting score would be 3. Samples with higher forgetting scores are more challenging for the model to learn and retain. To form the coreset, the method ranks all samples by their forgetting scores and selects the top $k$ samples with the highest scores:

$$C = \{(x_i, y_i) \mid x_i \text{ ranks among the top } k \text{ in high forgetting score}\} \tag{18}$$

This approach ensures the coreset contains the most challenging and informative examples, which can help the model learn and retain knowledge more effectively.

**GraphCut** coreset selection models the dataset as a graph to identify a subset of representative data points by leveraging the relationships among samples. In this approach, each data points $x_i$ is represented as a node $v_i \in V$, and each edge $e_{ij} \in E$ between nodes $v_i$ and $v_j$ represent weighted by a similarity metric $s(x_i, x_j)$ in a graph $G = (V, E)$. A similarity metric $s(x_i, x_j)$ with a scaling factor $\sigma$ can be defined as:

$$s(x_i, x_j) = \exp\left(-\frac{|x_i - x_j|^2}{2\sigma^2}\right), \tag{19}$$

The goal is to form a coreset $C$ by selecting a subset of nodes such that the cut value of the partition is minimized while preserving representativeness. The cut value measures the total similarity between the coreset and the remaining dataset, or is defined formally as the sum of the weights of the edges crossing between the selected subset $C$ and the remaining nodes $V \setminus C$ as in Eq. 20. A lower cut value ensures that the selected coreset is less redundant and has minimal overlap with the rest of the dataset. Therefore, the final coreset $C$ can be obtained by using Eq. 21 where $k$ is the desired size of the coreset.

$$\text{Cut}(C, V \setminus C) = \sum_{i \in C, j \in V \setminus C} s(x_i, x_j). \tag{20}$$

$$C = \arg \min_{|C| \leq k} \text{Cut}(C, V \setminus C), \tag{21}$$

### A.3 Implementation Details

**Backbones.** To offer a more comprehensive evaluation, we test both from scratch and pretrained models across two architectures: ResNet18 (He et al., 2016) and Vision Transformer (ViT) (Dosovitskiy et al., 2021). In **ResNet18** trained from scratch, we observe how well it can learn task-specific features directly from the dataset. In contrast, the pretrained models **Pretrained-ResNet18** and **Pretrained-ViT** are initialized with ImageNet weights, giving them prior knowledge of visual patterns and structures, which helps them start with a robust foundation for CL.

**Metrics.** We utilize average accuracy (ACC) which measures the final accuracy averaged over all tasks and can be formulated as $ACC = \frac{1}{T} \sum_{i=1}^{T} A_{T,i}$ where $A_{T,i}$ represents the testing accuracy of task $T$ after learning task $i$. To observe learning-forgetting dynamics more in detail, we utilize heatmaps that show the accuracy of each task after every learning session instead of sharing a single numerical value.

### A.4 Results for Split-CIFAR100 with ResNet18

Figure A illustrates the accuracy results on the Split-CIFAR100 dataset after each task, comparing various class-incremental learning methods. For each method, we evaluate performance using both the full dataset and the best-performing coreset, chosen based on size and selection criteria optimal for each approach. Notably, the results show a pattern of improved accuracy when coresets are used, which aligns with observations made in the Split-CIFAR10 experiments. This accuracy boost can primarily be attributed to reduced forgetting, as training on a selected subset allows the model to retain important task information with less interference from previous tasks. Minimizing redundancy and focusing on coreset samples, provides a more targeted training approach and enhances overall model performance in class-incremental scenarios.

### A.5 Results for Pretrained Resnet18 and ViT

We also explore the effects of learning from high-value samples when prior knowledge is available through a pretrained backbone network. Although pretraining often provides a useful foundation, it does not have to consistently yield performance gains, as pretrained parameters are subject to continual fine-tuning with each new task. Our experiments on Split-CIFAR10 and Split-CIFAR100 datasets consistently demonstrate that learning from coreset samples improves incremental performance when using ImageNet pretrained ResNet18 and ViT, aligning with our previous findings (Table A, Table B, and Table C).

In experiments with ViT, we further validate the efficacy of coreset selection by incorporating CODA-Prompt, a prompt-based technique tailored to transformer architectures. The application of CODA-Prompt demonstrates that coreset selection remains effective within prompt-based frameworks. Together, these results suggest that coreset selection is a valuable strategy for enhancing class-incremental learning.

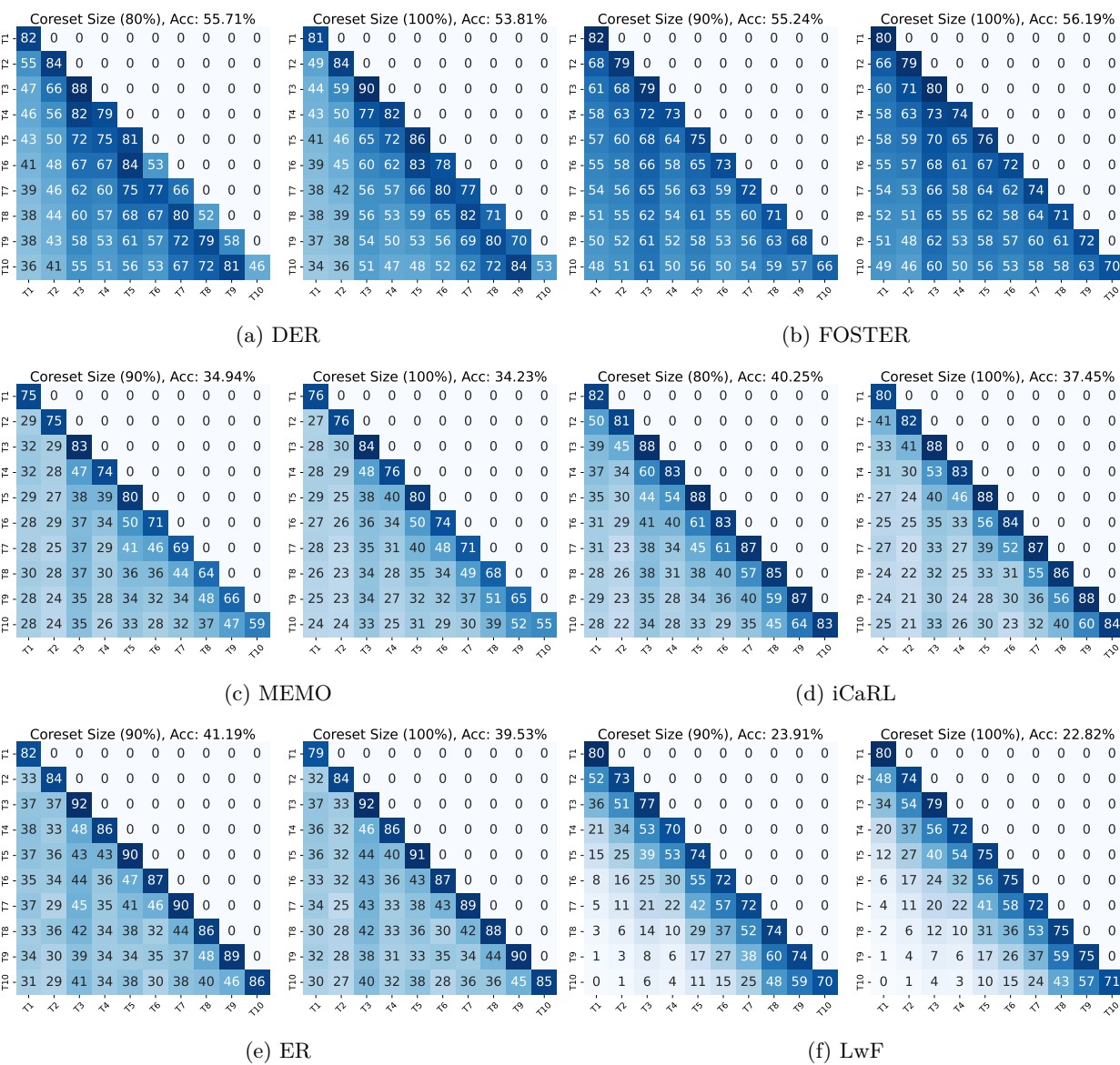

(a) DER

(b) FOSTER

(c) MEMO

(d) iCaRL

(e) ER

(f) LwF

Figure A: Accuracy [%] of each task after every learning session on different class-incremental learning methods with Split-CIFAR100. Its results align with Split-CIFAR10 and again incremental performance improves due to better knowledge retention.

Table A: Accuracy [%] on **Split-CIFAR10** with an ImageNet **pretrained ResNet18** shows that training with coreset samples improves incremental performance.

| | Fraction | 10% | 20% | 50% | 80% | 90% | 100% |
|---|---|---|---|---|---|---|---|
| | Random | 40.18 ± 5.28 | 53.93 ± 3.36 | 61.35 ± 2.37 | 66.66 ± 2.36 | 67.07 ± 2.51 | 67.85 ± 3.30 |
| | Herding | 57.35 ± 0.45 | 61.48 ± 1.32 | 65.84 ± 2.66 | 68.68 ± 3.74 | **71.36 ± 1.48** | 67.85 ± 3.30 |
| DER | Uncertainty | 61.23 ± 0.14 | 63.38 ± 0.40 | 67.63 ± 1.37 | 70.75 ± 2.71 | 70.92 ± 2.08 | 67.85 ± 3.30 |
| | Forgetting | 61.00 ± 0.23 | 65.02 ± 0.66 | 67.86 ± 1.84 | 71.72 ± 2.08 | 69.67 ± 2.78 | 67.85 ± 3.30 |
| | GraphCut | 62.00 ± 2.03 | 64.87 ± 1.92 | 68.39 ± 0.98 | 71.72 ± 1.65 | 71.19 ± 2.77 | 67.85 ± 3.30 |
| | Random | 42.82 ± 7.84 | 46.24 ± 2.57 | 60.15 ± 2.88 | 57.89 ± 4.07 | 58.32 ± 5.71 | 57.85 ± 3.09 |
| | Herding | 48.72 ± 4.27 | 50.35 ± 2.35 | 54.76 ± 4.46 | 56.71 ± 2.53 | 57.06 ± 3.39 | 57.85 ± 3.09 |
| FOSTER | Uncertainty | 54.51 ± 1.48 | 58.51 ± 2.97 | 58.34 ± 3.54 | 56.85 ± 4.81 | 56.35 ± 3.38 | 57.85 ± 3.09 |
| | Forgetting | 52.26 ± 0.45 | 55.52 ± 5.48 | 57.61 ± 3.53 | 57.65 ± 2.90 | 55.98 ± 2.98 | 57.85 ± 3.09 |
| | GraphCut | 53.84 ± 3.70 | **59.27 ± 3.28** | 58.04 ± 3.86 | 57.57 ± 3.71 | 56.09 ± 2.59 | 57.85 ± 3.09 |
| | Random | 37.49 ± 4.08 | 43.77 ± 10.63 | 48.74 ± 7.63 | 53.90 ± 2.21 | 59.34 ± 4.88 | 55.65 ± 8.06 |
| | Herding | 34.50 ± 7.48 | 44.94 ± 12.11 | 55.14 ± 7.53 | **62.84 ± 5.82** | 61.34 ± 5.19 | 55.65 ± 8.06 |
| MEMO | Uncertainty | 43.02 ± 5.27 | 50.06 ± 6.13 | 54.55 ± 6.44 | 61.21 ± 5.79 | 62.00 ± 5.72 | 55.65 ± 8.06 |
| | Forgetting | 37.64 ± 4.28 | 49.77 ± 8.80 | 54.98 ± 6.70 | 62.84 ± 5.78 | 61.84 ± 6.93 | 55.65 ± 8.06 |
| | GraphCut | 47.23 ± 3.19 | 52.04 ± 8.08 | 55.96 ± 6.87 | 61.57 ± 5.18 | 61.37 ± 5.61 | 55.65 ± 8.06 |
| | Random | 38.85 ± 0.13 | 47.22 ± 7.77 | 48.32 ± 3.87 | 48.97 ± 3.02 | 52.03 ± 5.62 | 53.37 ± 5.94 |
| | Herding | 53.52 ± 2.71 | 55.21 ± 1.45 | 53.68 ± 6.33 | 55.42 ± 5.13 | 55.38 ± 4.59 | 53.37 ± 5.94 |
| iCaRL | Uncertainty | 53.72 ± 3.14 | 56.03 ± 1.67 | 52.81 ± 5.12 | 56.82 ± 6.18 | 54.73 ± 5.88 | 53.37 ± 5.94 |
| | Forgetting | 53.20 ± 0.90 | 56.00 ± 4.88 | 54.76 ± 5.06 | 55.62 ± 5.33 | 54.98 ± 6.39 | 53.37 ± 5.94 |
| | GraphCut | **57.99 ± 2.41** | 57.98 ± 3.45 | 57.03 ± 3.85 | 55.63 ± 4.50 | 57.79 ± 5.47 | 53.37 ± 5.94 |
| | Random | 41.21 ± 2.43 | 43.55 ± 6.68 | 43.21 ± 5.02 | 44.16 ± 6.60 | 44.56 ± 6.71 | 45.01 ± 5.56 |
| | Herding | 38.28 ± 4.17 | 41.91 ± 3.25 | 47.91 ± 2.85 | 44.76 ± 7.06 | 43.17 ± 6.39 | 45.01 ± 5.56 |
| ER | Uncertainty | 36.23 ± 3.22 | 40.28 ± 7.42 | 42.19 ± 6.85 | 44.01 ± 8.18 | 43.81 ± 5.51 | 45.01 ± 5.56 |
| | Forgetting | 34.70 ± 3.03 | 42.90 ± 5.67 | 44.66 ± 6.07 | 44.41 ± 6.35 | 43.95 ± 6.01 | 45.01 ± 5.56 |
| | GraphCut | **52.26 ± 3.93** | 50.82 ± 4.91 | 46.33 ± 5.23 | 44.35 ± 7.20 | 45.11 ± 7.77 | 45.01 ± 5.56 |
| | Random | 30.80 ± 1.42 | 41.67 ± 1.89 | 45.95 ± 3.11 | 51.04 ± 0.38 | **54.74 ± 0.44** | 53.94 ± 0.79 |
| | Herding | 17.65 ± 0.23 | 21.74 ± 3.19 | 26.41 ± 3.72 | 29.85 ± 6.65 | 31.53 ± 6.01 | 53.94 ± 0.79 |
| LwF | Uncertainty | 25.21 ± 5.02 | 26.38 ± 6.01 | 27.76 ± 6.16 | 30.68 ± 6.37 | 32.13 ± 6.92 | 53.94 ± 0.79 |
| | Forgetting | 23.68 ± 1.81 | 26.99 ± 5.19 | 27.60 ± 5.33 | 30.74 ± 5.98 | 30.82 ± 6.81 | 53.94 ± 0.79 |
| | GraphCut | 26.45 ± 5.28 | 25.23 ± 4.16 | 27.79 ± 5.35 | 31.05 ± 5.38 | 31.78 ± 5.26 | 53.94 ± 0.79 |

Table B: Accuracy [%] on **Split-CIFAR100** with ImageNet **pretrained ResNet18**. Training with coreset samples improves the incremental performance also with a pretrained backbone.

| | Fraction | 10% | 20% | 50% | 80% | 90% | 100% |
|---|---|---|---|---|---|---|---|
| | Random | 20.38 ± 3.27 | 30.82 ± 0.76 | 44.96 ± 0.28 | 53.41 ± 1.96 | 52.23 ± 0.84 | 55.85 ± 0.38 |
| | Herding | 16.33 ± 4.78 | 22.13 ± 8.92 | 47.52 ± 2.47 | 55.51 ± 0.89 | 56.74 ± 1.09 | 55.85 ± 0.38 |
| DER | Uncertainty | 30.03 ± 0.62 | 40.53 ± 0.98 | 52.21 ± 0.78 | 56.94 ± 0.97 | **57.22 ± 0.59** | 55.85 ± 0.38 |
| | Forgetting | 30.08 ± 4.11 | 37.48 ± 5.50 | 51.88 ± 0.81 | 56.18 ± 1.53 | 56.16 ± 1.08 | 55.85 ± 0.38 |
| | GraphCut | 28.20 ± 1.64 | 38.79 ± 1.66 | 50.94 ± 1.59 | 55.76 ± 0.68 | 56.95 ± 1.77 | 55.85 ± 0.38 |
| | Random | 16.25 ± 0.27 | 19.71 ± 0.45 | 34.21 ± 3.55 | 50.80 ± 0.07 | 50.65 ± 1.36 | 56.63 ± 1.11 |
| | Herding | 12.51 ± 0.03 | 17.86 ± 1.39 | 37.88 ± 1.58 | 54.25 ± 2.37 | 55.40 ± 2.13 | 56.63 ± 1.11 |
| FOSTER | Uncertainty | 14.87 ± 1.03 | 23.91 ± 0.86 | 45.93 ± 1.50 | 55.21 ± 2.26 | **56.65 ± 2.27** | 56.63 ± 1.11 |
| | Forgetting | 18.44 ± 0.72 | 24.46 ± 1.84 | 44.04 ± 0.33 | 55.45 ± 2.08 | 56.30 ± 1.21 | 56.63 ± 1.11 |
| | GraphCut | 17.87 ± 2.30 | 22.10 ± 3.81 | 44.94 ± 0.94 | 55.51 ± 1.93 | 56.60 ± 2.16 | 56.63 ± 1.11 |
| | Random | 17.21 ± 1.91 | 25.29 ± 0.42 | 38.54 ± 3.05 | 43.16 ± 2.88 | 46.32 ± 3.75 | 46.70 ± 3.64 |
| | Herding | 10.94 ± 0.72 | 20.13 ± 0.21 | 36.26 ± 0.94 | 44.29 ± 0.75 | **46.87 ± 0.24** | 46.70 ± 3.64 |
| MEMO | Uncertainty | 17.85 ± 1.05 | 24.54 ± 0.15 | 37.92 ± 0.73 | 44.87 ± 0.30 | 46.10 ± 0.57 | 46.70 ± 3.64 |
| | Forgetting | 21.56 ± 0.52 | 28.20 ± 0.51 | 38.59 ± 1.06 | 44.49 ± 0.88 | 45.86 ± 0.58 | 46.70 ± 3.64 |
| | GraphCut | 27.60 ± 5.53 | 33.44 ± 4.45 | 40.38 ± 0.13 | 44.54 ± 0.29 | 45.60 ± 0.08 | 46.70 ± 3.64 |
| | Random | 20.09 ± 0.72 | 22.25 ± 0.93 | 30.08 ± 0.04 | 30.40 ± 1.16 | 33.60 ± 0.66 | 32.90 ± 0.80 |
| | Herding | 18.46 ± 0.72 | 24.80 ± 1.56 | 32.74 ± 2.12 | 34.70 ± 2.10 | 34.74 ± 2.08 | 32.90 ± 0.80 |
| iCaRL | Uncertainty | 22.70 ± 0.23 | 27.82 ± 0.88 | 32.68 ± 1.42 | 33.44 ± 1.26 | 34.04 ± 1.65 | 32.90 ± 0.80 |
| | Forgetting | 24.22 ± 0.69 | 30.00 ± 1.38 | 33.85 ± 2.05 | 34.16 ± 2.72 | 35.21 ± 2.10 | 32.90 ± 0.80 |
| | GraphCut | 28.88 ± 0.34 | 30.93 ± 2.39 | **35.40 ± 1.56** | 34.17 ± 0.96 | 34.02 ± 1.47 | 32.90 ± 0.80 |
| | Random | 16.6 ± 3.59 | 22.35 ± 0.04 | 26.09 ± 0.34 | 25.42 ± 0.10 | 24.91 ± 0.16 | 24.58 ± 0.46 |
| | Herding | 15.2 ± 0.8 | 19.9 ± 0.32 | 25.16 ± 0.97 | 25.94 ± 1.52 | 25.30 ± 0.83 | 24.58 ± 0.46 |
| ER | Uncertainty | 14.4 ± 0.46 | 17.56 ± 0.62 | 22.78 ± 0.24 | 24.04 ± 0.14 | 25.58 ± 0.61 | 24.58 ± 0.46 |
| | Forgetting | 19.01 ± 0.63 | 21.72 ± 0.14 | 25.57 ± 0.69 | 25.69 ± 0.89 | 26.26 ± 1.55 | 24.58 ± 0.46 |
| | GraphCut | 27.01 ± 0.34 | **28.99 ± 1.63** | 27.52 ± 0.57 | 26.03 ± 1.43 | 25.43 ± 0.86 | 24.58 ± 0.46 |
| | Random | 10.39 ± 0.36 | 12.63 ± 1.40 | 20.69 ± 0.70 | 22.78 ± 0.38 | **25.01 ± 0.46** | 24.31 ± 0.57 |
| | Herding | 4.15 ± 0.11 | 5.44 ± 0.10 | 9.47 ± 0.84 | 13.11 ± 1.53 | 13.77 ± 0.96 | 24.31 ± 0.57 |
| LwF | Uncertainty | 7.42 ± 0.01 | 9.15 ± 0.22 | 11.00 ± 0.58 | 13.29 ± 1.18 | 14.46 ± 0.99 | 24.31 ± 0.57 |
| | Forgetting | 7.26 ± 0.24 | 8.22 ± 0.17 | 10.89 ± 0.86 | 13.06 ± 1.14 | 14.04 ± 0.94 | 24.31 ± 0.57 |
| | GraphCut | 6.59 ± 0.32 | 7.23 ± 0.32 | 11.13 ± 0.67 | 13.21 ± 1.21 | 13.65 ± 1.13 | 24.31 ± 0.57 |

Table C: Accuracy [%] on **Split-CIFAR100** with ImageNet **pretrained ViT**. Training with coreset samples improves the incremental performance also with a pretrained backbone.

| | Fraction | 10% | 20% | 50% | 80% | 90% | 100% |
|---|---|---|---|---|---|---|---|
| | Random | 61.51 ± 0.36 | 61.88 ± 1.00 | 64.39 ± 0.78 | 63.12 ± 0.02 | 64.10 ± 1.22 | 60.83 ± 1.93 |
| | Herding | 68.25 ± 1.44 | 69.26 ± 1.15 | 70.07 ± 0.15 | 68.58 ± 1.03 | 68.88 ± 1.92 | 60.83 ± 1.93 |
| DER | Uncertainty | **74.44 ± 0.37** | 71.30 ± 0.42 | 69.68 ± 0.16 | 68.92 ± 0.37 | 70.28 ± 0.18 | 60.83 ± 1.93 |
| | Forgetting | 70.70 ± 2.70 | 73.10 ± 0.55 | 69.92 ± 1.23 | 68.15 ± 0.63 | 68.05 ± 0.09 | 60.83 ± 1.93 |
| | GraphCut | 72.58 ± 0.27 | 72.29 ± 0.03 | 69.70 ± 1.58 | 68.88 ± 1.47 | 68.37 ± 2.21 | 60.83 ± 1.93 |
| | Random | 72.51 ± 2.67 | 81.41 ± 0.67 | 84.97 ± 0.56 | 85.91 ± 0.28 | 86.35 ± 0.42 | 86.74 ± 0.30 |
| | Herding | 68.84 ± 0.01 | 78.87 ± 0.34 | 83.68 ± 0.23 | 85.41 ± 0.34 | 85.58 ± 0.27 | 86.74 ± 0.30 |
| FOSTER | Uncertainty | 77.10 ± 0.59 | 82.68 ± 0.26 | 85.17 ± 0.23 | 86.03 ± 0.12 | 85.83 ± 0.19 | 86.74 ± 0.30 |
| | Forgetting | 77.00 ± 1.53 | 82.61 ± 0.30 | 84.90 ± 0.39 | 85.74 ± 0.33 | 86.03 ± 0.24 | 86.74 ± 0.30 |
| | GraphCut | 74.64 ± 0.79 | 79.72 ± 0.42 | 84.14 ± 0.08 | 85.09 ± 0.13 | 85.68 ± 0.41 | 86.74 ± 0.30 |
| | Random | 14.84 ± 0.20 | 17.87 ± 0.90 | 23.74 ± 5.85 | 27.24 ± 5.37 | 30.07 ± 7.65 | 36.12 ± 0.16 |
| | Herding | 27.79 ± 1.15 | 24.68 ± 1.79 | 28.22 ± 2.03 | 31.02 ± 0.66 | 30.07 ± 0.45 | 36.12 ± 0.16 |
| MEMO | Uncertainty | 29.21 ± 1.47 | 29.34 ± 1.07 | 32.13 ± 0.76 | 31.88 ± 2.99 | 30.95 ± 0.10 | 36.12 ± 0.16 |
| | Forgetting | 35.14 ± 1.79 | 31.72 ± 0.71 | 29.29 ± 1.46 | 31.47 ± 2.11 | 31.00 ± 2.94 | 36.12 ± 0.16 |
| | GraphCut | 33.74 ± 1.66 | 32.46 ± 2.07 | 33.45 ± 3.05 | 30.67 ± 3.23 | 28.38 ± 2.63 | 36.12 ± 0.16 |
| | Random | 71.24 ± 1.50 | 71.79 ± 2.62 | 70.62 ± 1.56 | 68.30 ± 1.72 | 68.79 ± 2.38 | 66.03 ± 0.61 |
| | Herding | 68.34 ± 0.21 | 69.85 ± 0.25 | 71.11 ± 0.48 | 70.72 ± 0.41 | 69.09 ± 0.59 | 66.03 ± 0.61 |
| iCaRL | Uncertainty | 74.88 ± 0.41 | 74.11 ± 0.14 | 70.61 ± 0.13 | 70.99 ± 0.34 | 69.20 ± 0.45 | 66.03 ± 0.61 |
| | Forgetting | 73.21 ± 0.58 | 73.51 ± 0.40 | 71.91 ± 0.76 | 70.74 ± 0.23 | 70.61 ± 1.03 | 66.03 ± 0.61 |
| | GraphCut | 72.74 ± 4.08 | **73.68 ± 1.78** | 71.72 ± 0.52 | 73.05 ± 2.42 | 73.59 ± 2.28 | 66.03 ± 0.61 |
| | Random | 69.52 ± 2.83 | 73.54 ± 1.81 | 73.59 ± 0.10 | 73.36 ± 0.16 | 72.39 ± 0.52 | 67.95 ± 0.86 |
| | Herding | 67.47 ± 1.53 | 70.57 ± 0.20 | 71.43 ± 1.43 | 72.65 ± 0.60 | 72.24 ± 0.16 | 67.95 ± 0.86 |
| ER | Uncertainty | 73.97 ± 0.25 | 72.71 ± 1.94 | 71.68 ± 0.38 | 72.68 ± 0.84 | 70.31 ± 0.37 | 67.95 ± 0.86 |
| | Forgetting | 71.32 ± 0.73 | 71.31 ± 0.24 | 71.50 ± 1.06 | 72.00 ± 0.45 | 72.09 ± 0.27 | 67.95 ± 0.86 |
| | GraphCut | **76.59 ± 0.35** | 76.39 ± 1.68 | 74.87 ± 0.46 | 70.09 ± 0.25 | 70.69 ± 0.66 | 67.95 ± 0.86 |
| | Random | 52.76 ± 2.27 | 60.26 ± 2.62 | 64.73 ± 1.56 | 65.71 ± 0.70 | 65.35 ± 0.85 | 66.63 ± 1.41 |
| | Herding | 22.99 ± 0.13 | 24.44 ± 0.13 | 27.57 ± 0.49 | 29.46 ± 0.67 | 31.10 ± 0.40 | 66.63 ± 1.41 |
| LwF | Uncertainty | 25.17 ± 0.60 | 26.27 ± 0.31 | 28.78 ± 0.26 | 30.19 ± 0.64 | 30.31 ± 0.04 | 66.63 ± 1.41 |
| | Forgetting | 24.99 ± 0.29 | 26.50 ± 0.18 | 27.63 ± 0.74 | 31.22 ± 0.77 | 30.52 ± 0.44 | 66.63 ± 1.41 |
| | GraphCut | 23.32 ± 1.24 | 25.84 ± 0.98 | 29.53 ±1.47 | 29.66 ± 0.45 | 31.82 ± 0.75 | 66.63 ± 1.41 |
| | Random | 78.99 ± 1.42 | 81.62 ± 1.89 | 84.01 ± 0.11 | 84.64 ± 0.38 | 85.45 ± 0.44 | 85.37 ± 0.79 |
| | Herding | 73.21 ± 1.23 | 74.31 ± 1.19 | 83.21 ± 0.72 | 85.51 ± 0.65 | 85.73 ± 0.01 | 85.37 ± 0.79 |
| CODA-Prompt | Uncertainty | 78.48 ± 1.02 | 82.32 ± 1.01 | 85.20 ± 0.16 | 85.64 ± 0.37 | 85.57 ± 0.92 | 85.37 ± 0.79 |
| | Forgetting | 78.30 ± 1.81 | 82.48 ± 1.19 | 84.73 ± 0.33 | 85.73 ± 0.98 | 86.33 ± 0.81 | 85.37 ± 0.79 |
| | GraphCut | 80.55 ± 1.28 | 83.33 ± 1.16 | 84.31 ± 0.35 | 85.26 ± 0.38 | **86.34 ± 0.26** | 85.37 ± 0.79 |

