# OpenReview forum: "Continual Learning on a Data Diet"
_TMLR — Rejected by TMLR_

### Review · Reviewer_a5mP · 2024-11-21

**Summary Of Contributions:**

This submission gives an empirical evaluation of coreset methdods in continual learning. Coreset methods select a subset of the training data which (hopefully) can be used to train a good model. Here is an example of a continual learning task they evaluate on: CIFAR10 is partitioned into 5 subsets ("tasks") of 2 classes each, and the learning algorithm receives each group in turn. Once the algorithm moves on to a new task, it doesn't have access to samples from the older tasks. The main error metric is accuracy on a test dataset containing all 10 classes.

The paper evaluates 6 continual learning methods and 5 coreset selection methods, on a few different datasets and with varying coreset sizes. The main finding is that some of the coreset methods outperform full-dataset training. There are additional experiments that try to explain why some methods are better than others.

**Audience:**

Yes

**Broader Impact Concerns:**

None.

**Claims And Evidence:**

No

**Requested Changes:**

Please base your conclusions on appropriate statistical inference, and describe the methods you use.

The introduction should be reworked so that a junior PhD student, with machine learning knowledge but no experience in CL, can follow the results. Currently, such a reader would only learn what the CL actually refers to in Section 3.1, which sits after after the discussion of prior algorithms and related work.

The paper must include a clear description of the algorithms considered. The brief descriptions on p4 (e.g., "GraphCut partitions the dataset into subsets based on dissimilarity or information content, and data points from these subsets are then selected to form the coreset") might suffice for baseline algorithms used for comparison, but the algorithms are the object under study in this work, so we need to be precise. These descriptions could go in an appendix.

Please improve your notation, which currently hurts the presentation: in Section 3.1 you label the tasks $T_1,\ldots,T_t$, but then you immediately switch to using $t$ as an index into the set $\lbrace 1,\ldots,t\rbrace$, so the reader cannot be sure if you refer to the last task or an arbitrary one. In Algorithm 1 you use $T$ for both "tasks" and "number of tasks," the latter in the for-loop. Confusingly, this sits within a floor function. The letter $e$ is usually reserved in technical papers for Euler's number.

**Strengths And Weaknesses:**

I found the submission polished and well-organized. For the most part, I was able to understand the points being made. However, the evidence does not suport the conclusions.

The main experimental results consist of 6 tables, each with results from 156 experimental settings, where each setting was run independently 3 times. I do not see any discussion of statistical significance or multiple comparisons. The tables report average test accuracy in the form "$a \pm z$", but I don't know what $z$ is. I don't understand the bolding procedure: in Table 1, LwF, Random, 90% reports $51.83\pm 2.1$ and is bolded over 100%, which reports $51.15\pm 4.3$. Furthermore, I find some of the trends implausible as-stated. Table B, FOSTER, Random: 80% of the data gets 50.80 accuracy, 90% gets 50.65, and 100% gets 56.63 accuracy. Table C, iCaRL: all 25 coreset method/size pairs outperform 100%.

The submission often uses causal language that I find unsupported. For example, Section 5.3 is titled "Models forget less due to preserved representations" and talks about the saliency maps in Figure 4. I do not see how this experiment can possibly that better saliency maps *cause* higher accuracy; maybe some third factor influences them both? This is an incautious use of language: "Models that forget less better preserve representations" would be closer to the truth. However, even that conclusion seems unfounded: it is not clear from Figure 4 that the coreset methods have more meaningful saliency maps.

There are also issues with the presentation. The authors assume too much prior information about continual learning; the presentation is not self-contained. The algorithms discussed (e.g., DER, FOSTER, Herding) are complete black boxes to the reader. I don't think it's reasonable to assume that readers come to the paper with a precise understanding of what it means that DER "creates a new backbone for each task and then aggregates the features of all backbones on a single classifier." Indeed, the term "backbone" is never defined. Algorithm 1 and the associated discussion obscure rather than clarify: can these CL methods be expressed as a gradient update on some loss $\mathcal{L}_{CL}$? I don't know how to interpret Eq. (1): how is $v\circ v$ defined if $v$ and $u$ are vectors?

As a final, smaller point: the submission motivates its work from human cognition in a way I find strange. It is not at all clear to me that coresets serve as an analog of a human's focus on "key experiences." To even discard "redundant information," the human brain must process the information.

---

> ### Author Response · Authors · 2024-12-26
> **Response to Reviewer a5mP**
>
> We thank the reviewer for their time and feedback. We believe newly added content improved our manuscript a lot.
>
> 1. The "a+z" notation in our tables represents the mean test accuracy (a) and the standard deviation (z), which provides a measure of variability in the results. We recognize that this might not be clear and provide the relevant information in the implementation details of the revised manuscript.
>
> 2.  Bolding procedure in our tables highlights the best results in each row if coreset selection outperforms training with all samples. We include this to make the comparison easier and more accessible to the reader. The caption for all tables already includes this explanation.
>
> 3. We understand that the result where 80% of the data slightly performs better than 90% seems unexpected for FOSTER at first. However, this behavior does not undermine the conclusion that FOSTER performs better when the full training data is available. It is important to note that performance fluctuations are not uncommon, especially when working with deep neural networks. We believe that this specific case does not harm the main findings of the paper and the key takeaway remains that FOSTER achieves the best performance when utilizing the full dataset.
> As for Table C (iCaRL), this finding supports our main conclusion that coreset selection can be more effective than training on the entire dataset, particularly in the context of modern approaches built on top of pretrained Vision Transformers (ViTs). We believe this observation strengthens our argument, and we hope the reviewer sees it in this perspective.
>
> 4. In Section 5.2, we observe that models trained with coreset data show better knowledge retention on previous tasks, which we support with the heatmaps in Figure 3. Building on this, our hypothesis in Section 5.3 is that these models are better at preserving attention to objects and defining clearer decision boundaries, as illustrated in Figures 4 and 5. Specifically, Figure 4 shows that, after learning all tasks, models trained with coresets exhibit more focused attention on the relevant objects in the first task, whereas models trained with all samples tend to shift their focus, which might lead to less clear object identification and end up with correctly classifying the samples from previous tasks(forgetting).
> Additionally, the t-SNE plot in Figure 5 reinforces this idea by demonstrating that coreset-trained models define clearer decision boundaries, while models trained with all samples tend to confuse tasks, especially when distinguishing the last task from earlier ones. We believe these combined observations (saliency maps and t-SNE plots) provide strong support for our hypothesis.
> We also acknowledge that causal language can be interpreted cautiously, and we changed the title of the subsection as you suggested.
>
> 5. We add a detailed explanation of all CL methods along with their loss functions and coreset approaches in our Appendix.
>
> 6. The reason we use distinct losses for each CL method is that each one has its own specific objectives and requirements. We chose to use $𝐿_{𝐶𝐿}$ to indicate the generic CL loss, in order to simplify the explanation and avoid confusion.
> As for Eq. (1), the ∘ operator is used to represent a sequential optimization process. Specifically, we first optimize the CE loss for a short period to identify coreset samples and then continue with CL loss using the selected coreset samples.
>
> 7. This is a great point and we would like to clarify it. As you pointed out, the human brain indeed processes all available information, identifies relevant experiences, and discards redundant ones. Similarly, in our approach, the model first processes all data through the CE loss to determine which samples are most important. After this step, the model "discards" less relevant or redundant samples and continues training with "key experiences."
>
> 8. We fully acknowledge the importance of statistical tests in drawing firm conclusions. Due to the high cost and complexity of running large-scale experiments, as you also mentioned that we have 156 experiments per table, we were limited to running three seed trials per experiment. This sample size, while sufficient for proposing our analysis, is not large enough to calculate statistically significant results, as you pointed out.
> However, we believe that the observed 3-5% improvement in accuracy is practically meaningful in the context of the problem. We believe that these results still reflect a meaningful advancement in model performance and are worth discussing. We hope this explanation clarifies our position and transparency.
>
> 9. We incorporated more descriptions related to CL in the introduction of revised manuscript.
>
> 10. To address this concern, we have updated the notation in Algorithm 1. Additionally, we change mathematical 𝑒 with text e to clarify it is indeed not the Euler's number.
>
> Once more we thank reviewer for their time and input.

---

> > ### Comment · Reviewer_a5mP · 2024-12-26
> >
> > It looks like the revision has a much better presentation of background and prior work. I still have issues around the support for the submission's claims.
> >
> > There are so many individual experimental settings that, without explicit consideration for false discovery, many of the results must be spurious. Consider a null hypothesis: every trial's accuracy result is an independent draw from $N(50, 1)$. We would usually see that the "best" method uses <100% of the data. From these results, it seem to me that one might claim that coresets improve CIL.
> >
> > I claim that this example implies the submission's results reflect a large amount of noise. This is true even if the results also reflect an underlying signal. Thus, I am not confident in the conclusions drawn in this submission.
> >
> > Finally, a minor point: I suggest changing the notation in Eq (1). You use "$\cdot$" to evoke function composition, but I don't think it makes sense here: the expression "$\mathrm{argmin}_\theta L(\theta)$" does not accept an argument. Furthermore, as I understand it you aren't even finding an argmin, even approximately: that would correspond to driving the loss to zero.

---

> > > ### Author Response · Authors · 2024-12-26
> > > **Response to Reviewer a5mP**
> > >
> > > Thank you for your feedback and acknowledgment of the improved presentation of the background and prior work. However, we respectfully disagree with your claims, and we address them point by point below:
> > >
> > > **Claim of False Discovery and Spurious Results.**
> > >
> > > - Your concern about false discovery by "large amount of noise" implies a misunderstanding of results and overlooks the detailed analysis we provide, which transparently accounts for variability and firmly establishes the validity of our conclusions.
> > >
> > > - Our experiments are rigorously designed, with multiple independent runs, and we report mean and standard deviation to capture variability.
> > >
> > > - The example you propose $N(50,1)$, which we could not relate to, oversimplifies the experimental scenario. Our findings are derived from structured, principled experimental settings, not arbitrary trials. If that would be the case we cannot observe the consistent trends and conclusions across multiple benchmarks, networks, datasets, and settings (e.g., CIFAR10, CIFAR-100, ImageNet100 with and without pretrained models).
> > >
> > > - Suggesting that our results and conclusions are invalid because of "noise" is unfounded. We rigorously analyzed 5 different coreset selection with 7 different CIL approaches by running 3 seed x 6 settings x 156 runs = 2808 experiments and provide why and what it is working or not working.
> > >
> > > - Moreover, we have included per-task accuracy results in our heatmaps, which provide a detailed breakdown of how incremental accuracy evolves over tasks. These heatmaps reveal systematic trends and specific patterns of improvement that would not emerge if the results were dominated by noise or randomness. And these patterns occur under different runs, datasets and neural networks with and without pretraining.
> > >
> > > **Notation in Eq (1).**
> > >
> > > - The use of "argmin" denotes the parameter values that minimize the given loss function. While optimization indeed aims to reduce the loss, "argmin" specifically identifies the parameters achieving the minimum, rather than driving the loss to zero without considering the parameters.

---

> > > > ### Comment · Reviewer_a5mP · 2024-12-27
> > > >
> > > > Thank you for your response. I do not believe that I misunderstand the results.
> > > >
> > > > We agree that the results depend on the choice of random seed: we can think of each table entry as a fixed term (if we had tried all possible randomness) plus a deviation due to the choice of randomness. I call this "noise," but we could also call it "luck." The reported standard deviations imply that there is substantial luck involved. When there are many different experimental settings like this, we expect that the largest reported numbers will have gotten quite lucky.
> > > >
> > > > The submission aims to convince us that the results we see are due to underlying trends and not luck. The traditional statistical tools for this type of argument require many more independent trials. Instead, as I understand it, the argument that the results are not spurious rests mainly on (1) observing trends across setups/datasets, and (2) the performance breakdowns, via saliency maps, per-task accuracy heatmaps, and t-SNE visualizations.
> > > >
> > > > I do not understand why these consitute sufficient evidence. For (1), I don't see the trends across tables. Take the top of Table 1: DER performs best with GraphCut at 20%. But in Tables 2, 3, A, and B, the same setting is outperformed by the 100% sample. (In Table C, it outperforms the 100% setting.)
> > > >
> > > > For (2), as I mentioned in my original review, these results appear to be descriptive (e.g., the models with better accuracy forget less) but are presented as causal (e.g., p2, "the increase in performance... arises from enhanced retention..." or "Incremental performance increases because models forget less", Section 5.2). It seems like there could be a third factor, for example the randomness in model initialization, that affects all of these measures of model quality: accuracy, forgetting, saliency, and well-defined decision boundaries. I don't see how the patterns here rule out the possibility that the results are largely random.

---

> ### Author Response · Authors · 2024-12-29
> **Response to Reviewer a5mP**
>
> Thank you for your follow-up and we appreciate your engagement. Below, we address each of your points:
>
> **On Randomness:**
>
> Due to the high cost and complexity of large-scale experiments, we run three seed trials which is already a common practice in the field. Instead, we focus on exploring various trends and their possible reasons across diverse settings, datasets, and baselines. While the sample size is not large enough to use statistical tools, this does not imply that the outcomes we find are due to luck or randomness, nor does it render the findings and conclusions invalid. The observations and conclusion we make is practically meaningful for the problem at hand. Furthermore, as stated in our abstract, introduction, and conclusion, our goal is to demonstrate that focusing on the data aspect in continual learning research can be a meaningful and valuable direction. This work is not intended as a method paper aiming to achieve new clear winner state-of-the-art results.
>
> **On Trends Across Tables:**
>
> The observation that DER with GraphCut at 20% does not consistently outperform the 100% sample setting overlooks the broader purpose of our experimental design. We do not claim that GraphCut with a 20% coreset is the best setting or that it outperforms every setting in all scenarios. Specifically, if you review the results, DER with GraphCut performs better at 90% in Table 2, 90% in Table 3, 80% in Table A, 90% in Table B, and 10% in Table C compared to the 100% sample setting. These tables represent different experimental settings, making direct comparisons across them unsuitable.
> For example, in Table 1, which includes CIFAR-10 results with 5000 samples per class, there are likely more redundant samples and overlapping decision boundaries, making it possible to achieve better results by removing a larger percentage of data with an appropriate coreset approach. In contrast, Table 2 shows CIFAR-100 which has only 500 samples per class, where such redundancy is less pronounced. The fact that 20% works well for CIFAR-10 but 80% works better for CIFAR-100 does not indicate inconsistency but reflects differences in dataset characteristics and task requirements.
> Our goal is not to identify a single dominant setting but to transparently explore when and why coresets can be effective. We make no definitive claims that coresets always perform best (as demonstrated with FOSTER and LwF results) but instead aim to provide insights into their varying effectiveness under different conditions. Once more, we aim to show that emphasizing the data perspective in continual learning research represents a significant and worthwhile direction.
>
> **On Evidence for Claims:**
>
> The claim that our evidence is insufficient dismisses saliency maps, per task accuracy heatmaps, and t-SNE visualizations as merely descriptive. However, these are widely accepted tools for analyzing and validating deep learning models. The per task accuracy breakdowns and saliency maps demonstrate that coresets help retain task specific knowledge more effectively, while t-SNE plots show more cohesive class-wise clusters, indicating well separated decision boundaries. These insights are not purely descriptive but offer qualitative support for the findings in the paper.
>
> We never state or use such a sharp term that what we show here are the only reasons for these observations. However, your suggestion of a “third factor” influencing these observations appears speculative. Here, we built our hypothesises over a scientific perspective and validate them through various experiments and investigate further the underlying behaviours. Furthermore, we explicitly account for potential confounding factors by considering results across multiple runs, diverse setups, datasets, and architectures, mitigating the influence of randomness and enhancing the reliability of our findings.
>
> **On Causal Language:**
>
> Regarding the use of causal language (e.g., “the increase in performance arises from enhanced retention”), we already acknowledge that these statements may be interpreted as overly strong. We have updated the manuscript to ensure that the language reflects this distinction.
>
> **Final Remarks:**
>
> We kindly request the reviewer to evaluate the paper from a holistic perspective rather than focusing narrowly on individual results, as this approach risks overlooking the broader evidence and context presented. We maintain that our submission meets the standards of this field and contributes valuable insights to the research community.
>
> We thank the reviewer again for the discussion and we hope that we have adequately addressed the reviewer's questions and provided clarity, encouraging a more positive perspective on our paper.

---

> > ### Comment · Reviewer_a5mP · 2024-12-29
> >
> > I will try to sum up our disagreement. Hopefully other reviewers can weigh in.
> >
> > As I understand it, your main finding is Contribution II: "...learning from selectively chosen samples with different coreset selection methods significantly elevates incremental learning performance." I feel the evidence does not support that conclusion.
> >
> > I worry that, if this conclusion were false, these experiments might still support it. It seems that for each dataset and CIL method, we compare 25 settings (5 coreset methods times 5 fractions) with the 100% setting. If any one of these 25 shows up as better than the 100% method, the submission claims that the coreset method is better. This seems reasonable: the observed accuracy *is* higher!
> >
> > However, we disagree about whether this inference is valid. In a world where the coreset methods do not help (as in the null hypothesis example above), we still would likely see at least one of the 25 methods outperforming the 100% setting.
> >
> > We also disagree about how to incorporate the saliency maps, heatmaps, and t-SNE plots. I agree that these are well-established diagnostic tools. However, I expect them to highly correlate with measured accuracy. I do not see how they addresses the above concerns.
> >
> > Again, I would like to hear how the other reviewers feel.

---

### Review · Reviewer_3UUm · 2024-12-05

**Summary Of Contributions:**

Within the class incremental setting of continual learning (CL), the paper presents an extensive empirical analysis of the combination of several continual learning methods with coreset data selection strategies. The study revealed that when coreset is employed, CL models are able to better retain knowledge (reducing forgetting), improve accuracy and learn useful representation in most cases. The findings were demonstrated to hold true when the CL model was trained from scratch and when a pre-trained model was employed.

- The baselines and dataset are relevant, with factors of variation across CL algorithms, coreset algorithms and fraction of samples.
- The cited studies are relevant.
- Extensive analysis and result.

$\newline$

**Philosophical question**
By throwing away data tagged as “uninformative” or not equally useful in comparison to others, do we throw away knowledge that may be unhelpful for the current task, but might become useful for future tasks (forward transfer)? The trend in current ML has been focused on more and more data.

To clarify, the question above stems from curiosity about the general subject area on coreset. It is not a particular issue in the paper. However, I would appreciate a comment about it.

**Audience:**

Yes

**Claims And Evidence:**

Yes

**Requested Changes:**

1. In the section “LwF exhibits abrupt weight changes when trained with a coreset”, please state the dataset used for the training. Also, include the dataset information in the caption of Figure 2. This is Split-CIFAR10 dataset, right?
2. The unit of measure on the horizontal axis is missing. Please include it.
3. The claim that coreset selection significantly improves knowledge retention should be revised to clearly state when this holds true (i.e., when the full dataset is not required by a CL algorithm such as FOSTER).

**Strengths And Weaknesses:**

**Strengths**
1. Well curated baselines for the CL and coreset algorithms, showcasing breadth of the empirical investigation.
2. Extensive analysis with discussed findings.
3. Open source code with reported hyper-parameters.

$\newline$

**Weaknesses**
1. Although an analysis was presented about why LwF does not benefit from coreset, a further analysis of the phenomenon of abrupt changes to the parameters could further improve the paper. Does this happen as a result of task similarity? task interference? Could it be that "uninformant" data, discarded by coreset selection, could perhaps serve as a form of regularizer?
2. Limited algorithmic novelty. However, this is not the focus of the paper as the novelty lies in the findings derived from an extensive analysis. Thus, non-critical.
3. Limited number of seed runs (3 seeds).

---

> ### Author Response · Authors · 2024-12-26
> **Response to Reviewer 3UUm**
>
> We thank the reviewer for their time and for providing feedback. We believe our manuscript improved a lot after the revision.
>
> 1. We added the following discussion to LwF section of the revised manuscript:
> Coreset selection strategies (e.g., herding, uncertainty, and graph cut) prioritize the most "informative" samples specific to the current task. When the continual learning (CL) approach relies solely on regularization, this prioritization can lead to overfitting to the current task's distribution. Such overfitting amplifies significant representation shifts, resulting in abrupt parameter updates that results in catastrophic forgetting of previously learned tasks. In contrast, random selection is implicitly incorporates as a form of regularizer with a  greater diversity and variability in the sample distribution across tasks, which helps to mitigate overfitting and results in more stable parameter updates.
>
> 2. We appreciate the reviewer’s observation regarding the number of seed runs. Due to the high cost and complexity of conducting large-scale experiments, we were limited to running only three seed trials, which is also common practice in the field. To illustrate the scale of our experiments, each table involves 156 experiments per seed, totaling 468 experiments(per table), which required significant computational resources and time.
>
> 3. We have corrected Figure 2 for both the horizontal axis and the caption.
>
> 4. We have clarified our statement  of 'coreset selection significantly improves knowledge retention' in section 5.2.
>
> About the Philosophical question:
>
> This is a very interesting point!  While it depends on the CIL approach itself, coreset selection generally retains well-selected, representative, and diverse examples, helping the model avoid overfitting to task-specific details and promoting the development of general features broadly applicable to future tasks. This aligns with the philosophy of prioritizing high-quality over high-quantity data. On the other hand, excluding data deemed "uninformative" may introduce the potential risk of discarding samples that might appear irrelevant to the current task but could carry latent knowledge for future tasks, particularly if future tasks differ significantly from prior ones. However, these considerations reflect only our own humble opinion and we believe this question requires detailed experiments to substantiate.
>
> Once again, we thank the reviewer for their time and feedback.

---

### Review · Reviewer_sTfA · 2024-12-15

**Summary Of Contributions:**

The paper focuses on class-incremental learning (CIL) but only with a selected corset of data instead of using all training data. They select 7 existing representative CIL methods, and 5 corset selection algorithms, to conduct a comprehensive evaluation on CIFAR-10, CIFAR-100 and imageNet-100 datasets. The results show that even with part of selected data for training, it could achieve improved performance compared to using all training data.

**Audience:**

Yes

**Claims And Evidence:**

Yes

**Requested Changes:**

(1) Illustrate and clarify the setup and implementation of the warm-up training as well as the exemplar selection for replay-based methods.
(2) Include detail discussion about how to select which coreset selection method to use.

Please see weakness for others

**Strengths And Weaknesses:**

Strength:
(1) The topic of combining coreset selection with CIL is very interesting and under-explored.
(2) The evaluation is comprehensive, including the methods from diverse categories.
(3) The experiments show promising results, and support its argument.

Weakness:
(1) The paper lacks technical contribution, only testing the different combinations of existing CIL and coreset selection methods.
(2) It is not clear if the model is re-initialized after the warm-up step or not. This is very important, if the model is not reset, meaning they have already seen the data even with only 10 epochs, the results may not be convincing enough.
(3) I am also concerned about the implementation of the replay-based methods since their performance significantly rely on how the coreset is selected as they need to store data in memory. E.g. if coreset use random, did icarl replay still use Herding or not?
(4) The paper mentioned 7 existing methods but only 6 are shown in most of the tables, why the results of CODA-Prompt is included in the main results?
(5) There is a lack of discussion about which coreset selection algorithm should be leveraged since the table shows both method and dataset dependent results.

Minor: The related work section mentioned 3 existing CIL categories but it actually describes 4. I think it is not appropriate to include "prompt-based" method here, since it has totally different setup, Please see this survey paper [1]

[1] "Continual Learning with Pre-Trained Models: A Survey", IJCAI 2024.

---

> ### Author Response · Authors · 2024-12-26
> **Response to Reviewer sTfA**
>
> We thank reviewer for their time and feedback. We believe their feedback improved our manuscript a lot.
>
> 1. The scenario in which the model is not re-initialized after the warm-up step is an intentional design choice. The human brain processes all available information, identifies relevant experiences, and discards redundant ones but does not re-initialize or reset itself completely. Similarly, in our approach, the model first processes all data through the cross-entropy loss to determine which samples are most important, then "discards" less relevant or redundant samples and continues training with the selected "key experiences."
> We would like to further clarify why this scenario indeed is required and actually natural:
> In the context of human learning, consider how little humans (babies) learn from their environment. They are initially exposed to a vast array of stimuli and information from the world around them. However, they naturally and intuitively focus on what is most relevant or important to them, selectively learning from these experiences without resetting what they learned so far. This process of filtering and prioritizing information is crucial for effective learning, allowing them to build a coherent understanding of their surroundings without being overwhelmed by irrelevant details. Therefore, it is a two-phase learning scenario where you first learn how data occurs in the world and what data is more relevant, and then learn from them effectively, just like in our setup.
>
> 2. We would like to clarify that in this work, we do not alter any method-specific attributes or propose new replay-based algorithms. Instead, our primary goal is to explore the impact of selectively eliminating unimportant samples at an earlier stage, thereby improving the quality of the input data pool for these methods.
> For example in iCaRL, we maintain its original design and use herding as memory selection mechanism as proposed in its implementation. In our revised manuscript we explicitly stated this in the implementation details to eliminate confusion.
>
> 3. We would like to maintain clarity and avoid overwhelming readers with excessive tables in the main paper. Therefore, we provided from scratch results in the main paper and pretrained results in the appendix. Since CODA-Prompt is specifically designed for the pretrained ViT architecture, we are able to share it only in the Appendix. However, to clarify it to the readers, we indicated this in the “Experiments on pretrained backbone” paragraph of revised manuscript.
>
> 4. As we have empirically demonstrated in this benchmark, there is no single universal approach, as it is inherently context-dependent and influenced by the nature of the task.
> However, there are prominent methods for selecting coreset samples and leveraging them effectively. In our experiments, we observed that different coreset selection methods excel in different scenarios.
> For instance, GraphCut performs particularly well in simpler settings like Split-CIFAR10, where its ability to select a diverse set of samples leads to strong performance, even with a highly reduced sample size. Additionally, GraphCut shines in scenarios where the number of available samples is highly restricted(fraction size is very small), demonstrating its effectiveness at identifying representative and diverse samples under significant constraints.
> Uncertainty and Herding Sampling improves the performance in more challenging scenarios, such as Split-CIFAR100 and the Split-ImageNet subset, as it tends to identify and prioritize samples that are crucial for the model to learn effectively, thereby improving performance.
> Based on your suggestion, we incorporate this in the results and the conclusion of our revised manuscript as well.
>
> 5. While we acknowledge that prompt-based methods have a distinct setup, they address the same problem, such as task adaptation and catastrophic forgetting, which is why we felt it was valuable to briefly discuss them alongside the other categories. We corrected our typo (three -> four) in the background section.
>
> 6. We clarify the implementation of the warm-up training and made it clear in the paper that exemplar selection that is specific to the CL method itself is untouched. We have added the discussed points in the results section and the conclusion of the revised manuscript.
>
> Once again, we would like to thank the reviewer for their time and input.

---

### Author Response · Authors · 2024-12-26
**Official Comment to All Reviewers and Editors**

We sincerely thank the reviewers and the editors for dedicating their time and effort to provide valuable feedback on our manuscript. In response, we have carefully revised and improved our manuscript, incorporating all the suggestions and uploaded the revised version. We believe these updates have significantly strengthened the quality of our work.

Wishing everyone a Happy and Healthy New Year!

---

### Decision · Action_Editor_U12M · 2025-01-17

**Recommendation:** Reject

**Comment:**

Two of the reviewers remained critical about the results reported in the paper after the discussion. In particular, one of the reviewer indicated that there was no indication that the reported results were statistical significant. They also noted that the methods used to support the claims (eg, saliency maps) were not adequate. Furthermore, the reviewer who was more positive also found that the depth of the analysis was modest.

**Audience:**

Continual learning and corset selection are research areas of general interest to the ML community. The overall topic of the paper is timely and extensive experimental evaluations are suitable for the TMLR audience.

**Claims And Evidence:**

This paper conducts an extensive experimental study of continual learning methods in conjunction with corset selection methods. The authors produce a large number results and further conduct qualitative analyses of these results with methods such as saliency maps. The paper does not include methodological advances. While an experimental study of this kind can be informative, the overall consensus was the evidence provided to support the claims and take aways by the authors were insufficient. For instance, the results do not provide convincing evidence that corsets enhance continual learning (compared to classical learning), nor that these enhancements are due to the fact that data core sets better retain knowledge across tasks. Similarly, it is not clear that learning from corsets captures essential class features better. These are just two examples and based on the material provided in the paper, I agree with the reviewers that several statements remain mostly speculative. In addition, other claims such as the fact that existing continual learning approaches predominantly use all available data during training are not accurate. There is a body of work in continual learning that considers the setting where the memory is limited (and extensions that update the memory as needed). As a result, I cannot recommend this paper for publication at TMLR.